# WHEN ATTENTION SINK EMERGES IN LANGUAGE MODELS: AN EMPIRICAL VIEW

**Xiangming Gu**[*1,2], **Tianyu Pang**[†1], **Chao Du**[1], **Qian Liu**[1], **Fengzhuo Zhang**[1,2],
**Cunxiao Du**[1], **Ye Wang**[†2], **Min Lin**[1]
[1]Sea AI Lab, Singapore    [2]National University of Singapore
{guxm, tianyupang, duchao, liuqian, zhangfz, ducx, linmin}@sea.com;
wangye@comp.nus.edu.sg

## ABSTRACT

Auto-regressive Language Models (LMs) assign significant attention to the first token, even if it is not semantically important, which is known as **attention sink**. This phenomenon has been widely adopted in applications such as streaming/long context generation, KV cache optimization, inference acceleration, model quantization, and others. Despite its widespread use, a deep understanding of attention sink in LMs is still lacking. In this work, we first demonstrate that attention sinks exist universally in auto-regressive LMs with various inputs, even in small models. Furthermore, attention sink is observed to emerge during the LM pre-training, motivating us to investigate how optimization, data distribution, loss function, and model architecture in LM pre-training influence its emergence. We highlight that attention sink emerges after effective optimization on sufficient training data. The sink position is highly correlated with the loss function and data distribution. Most importantly, we find that attention sink acts more like key biases, *storing extra attention scores*, which could be non-informative and not contribute to the value computation. We also observe that this phenomenon (at least partially) stems from tokens' inner dependence on attention scores as a result of softmax normalization. After relaxing such dependence by replacing softmax attention with other attention operations, such as sigmoid attention without normalization, attention sinks do not emerge in LMs up to 1B parameters. The code is available at https://github.com/sail-sg/Attention-Sink.

## 1 INTRODUCTION

Xiao et al. (2023b) showed that Large Language models (LLMs) allocate significant attention to the initial tokens, irrespective of their semantic relevance. This interesting phenomenon is termed as **attention sink** and has widespread applications, including streaming/long context generation (Xiao et al., 2023b; Han et al., 2024; Yang et al., 2024), KV cache optimization (Ge et al., 2023; Wan et al., 2024; Wu & Tu, 2024), efficient inference (Zhang et al., 2024b; Chen et al., 2024), model quantization (Liu et al., 2024b; Huang et al., 2024), and others.

A seminal of works attempted to understand attention sink. Among them, Cancedda (2024) clarified that attention sink primarily appears only on the first token. They attributed the phenomenon to the large norm of hidden states of the first token. This is referred to as *massive activations* (very few activations exhibit extremely large values compared to others) in Sun et al. (2024). Besides, Sun et al. (2024); Yu et al. (2024) observed that attention sink may also appear in several word tokens carrying limited semantic information and having no fixed position. Despite the above research efforts, a deep understanding of attention sink is still absent. Therefore, we conduct a comprehensive study to investigate when attention sink emerges. We defer full discussions on related work to Appendix A.

Based on open-sourced auto-regressive LMs, we show that the first token acts as biases: the angles between the first key and queries of other tokens are typically small, leading to attention sink. Then

---

[*]Work done during Xiangming Gu's internship at Sea AI Lab.
[†]Correspondence to Tianyu Pang and Ye Wang.

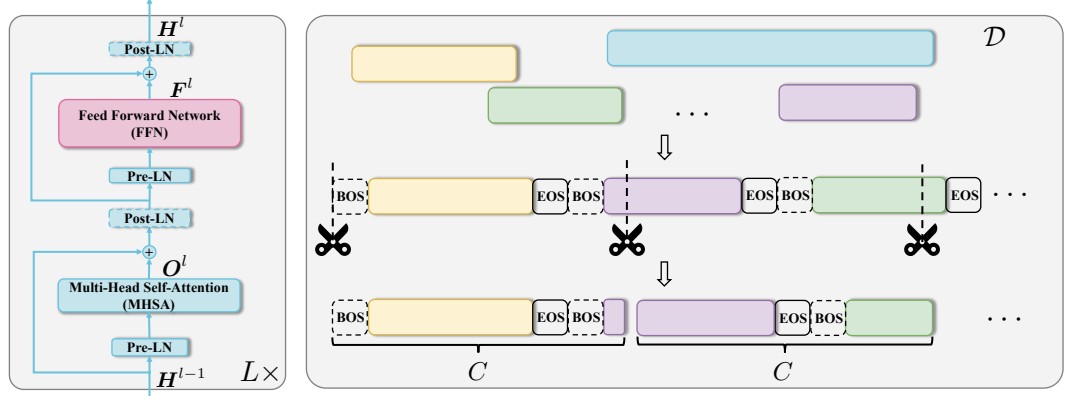

Figure 1: (*Left*) Architecture of pre-norm transformer block (we highlight the location of post-norm LN using dashed lines). We denote the output of MHSA as $\boldsymbol{O}^l$ and the output of FFN as $\boldsymbol{F}^l$. (*Right*) The packing strategy in the LM pre-training. All documents are concatenated with BOS (optional) and EOS tokens as the boundaries. Then it is chunked into equal-sized sequences with context length $C$.

we find that attention sink universally exists in auto-regressive LMs across different inputs, even in the small models or with random token sequences. Additionally, attention sink is observed to emerge during the LM pre-training before continual instruction tuning (Ouyang et al., 2022). This motivates us to focus on the LM pre-training, whose objective can be formulated as:

$$\min_{\theta} \mathbb{E}_{\boldsymbol{X} \sim p_{\text{data}}} \left[ \mathcal{L} \left( p_{\theta}(\boldsymbol{X}) \right) \right]. \tag{1}$$

In the remaining part of this paper, we investigate how the optimization (Section 4), data distribution (Section 5), loss function (Section 6), and model architecture (Section 7) influence the emergence of attention sink. We have the following conclusions:

- Attention sink emerges after LMs are trained effectively on sufficient training data. It appears less obvious in LMs trained with small learning rates. While weight decay encourages the emergence of attention sink.

- The sink position is highly related to the loss function and data distribution and can be shifted to other positions rather than the first token.

- Attention sink acts more like key biases, storing extra attention and meanwhile not contributing to the value computation. This phenomenon (at least partially) stems from tokens' inner dependence on attention scores due to the softmax normalization. After relaxing such dependence by replacing softmax attention with other attention operations, e.g., sigmoid attention without normalization, attention sinks do not emerge in LMs up to 1B parameters.

## 2 PRELIMINARIES ON LMS AND ATTENTION SINK

Let $f_{\theta}$ be an auto-regressive LM with $L$ transformer decoder blocks and $\boldsymbol{X} \in \mathbb{R}^{T \times |\mathbb{V}|} := \{\boldsymbol{x}_1, \boldsymbol{x}_2, \ldots, \boldsymbol{x}_T\}$ are the input tokens, where each token $\boldsymbol{x}_t$ is a one-hot encoding and $|\mathbb{V}|$ is the vocabulary size of tokenizer $\mathbb{V}$. The LM output is also a sequence $\boldsymbol{Y} \in \mathbb{R}^{T \times |\mathbb{V}|} := \{\boldsymbol{y}_1, \boldsymbol{y}_2, \ldots, \boldsymbol{y}_T\} = f_{\theta}(\boldsymbol{X})$, where $\boldsymbol{y}_t$ represents the predicted logits of $p(\boldsymbol{x}_{t+1}|\boldsymbol{x}_{\leq t})$.

**Transformer blocks.** In the forward pass, $\boldsymbol{X}$ is first embedded as $\boldsymbol{H}^0 \in \mathbb{R}^{T \times d} := \boldsymbol{X}\boldsymbol{W}_E + \boldsymbol{P}$, where $\boldsymbol{W}_E \in \mathbb{R}^{|\mathbb{V}| \times d}$ is the learnable word embedding, $\boldsymbol{P} \in \mathbb{R}^{T \times d}$ is the positional embedding, and $d$ is the hidden dimension. We denote $\boldsymbol{H}^l \in \mathbb{R}^{T \times d} := \{\boldsymbol{h}_1^l, \boldsymbol{h}_2^l, \ldots, \boldsymbol{h}_T^l\}, 1 \leq l \leq L$ to be the output of the $l$-th block. Each block comprises a multi-head self-attention (MHSA) operation and a feed-forward network (FFN). The block has either a pre-norm or post-norm structure according to the location of layer normalization (LN) (Ba et al., 2016; Zhang & Sennrich, 2019). Most of LLMs consider a pre-norm block, as also shown in Figure 1(*Left*):

$$\boldsymbol{H}^l = \text{FFN}(\text{LN}(\boldsymbol{O}^l + \boldsymbol{H}^{l-1})) + \boldsymbol{O}^l + \boldsymbol{H}^{l-1}, \ \boldsymbol{O}^l = \text{MHSA}(\text{LN}(\boldsymbol{H}^{l-1})), \tag{2}$$

while the post-norm transformer block is

$$\boldsymbol{H}^l = \text{LN}\left(\text{FFN}(\text{LN}(\boldsymbol{O}^l + \boldsymbol{H}^{l-1})) + \text{LN}(\boldsymbol{O}^l + \boldsymbol{H}^{l-1})\right), \ \boldsymbol{O}^l = \text{MHSA}(\boldsymbol{H}^{l-1}). \tag{3}$$

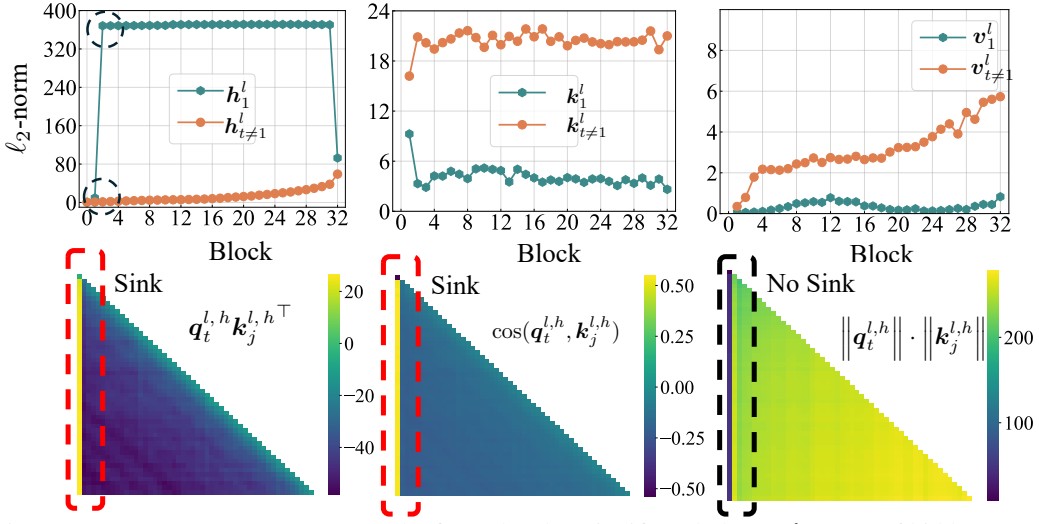

Figure 2: In LLaMA3-8B Base, (*Top*) the first token has significantly larger $\ell_2$-norm of hidden states, but much smaller $\ell_2$-norm of keys and values than the mean of other tokens; (*Bottom*) cosine similarity instead of norm product contributes to attention sink. We delay more visualizations to Appendix C.3.

**MHSA layers.** In the MHSA layer, the input $\boldsymbol{H}^{l-1}$ are first transformed into keys, queries, and values: $\boldsymbol{K}^{l,h} = \boldsymbol{H}^{l-1}\boldsymbol{W}_K^{l,h}, \boldsymbol{Q}^{l,h} = \boldsymbol{H}^{l-1}\boldsymbol{W}_Q^{l,h}, \boldsymbol{V}^{l,h} = \boldsymbol{H}^{l-1}\boldsymbol{W}_V^{l,h}$ for each head $1 \le h \le H$ (we omit the notation of LN when considering pre-norm design for simplicity). Here $\boldsymbol{W}_K^{l,h}, \boldsymbol{W}_Q^{l,h}, \boldsymbol{W}_V^{l,h} \in \mathbb{R}^{d \times d_h}, d_h = d/H$. Then the attention output is computed as

$$\boldsymbol{A}^{l,h} = \text{Softmax}\big(\boldsymbol{Q}^{l,h}\boldsymbol{K}^{l,h\top}/\sqrt{d_h} + \boldsymbol{M}\big), \quad \boldsymbol{O}^l = \text{Concat}_{h=1}^H \big(\boldsymbol{A}^{l,h}\boldsymbol{V}^{l,h}\big)\boldsymbol{W}_O^l, \tag{4}$$

where $\boldsymbol{M} \in \mathbb{R}^{T \times T}$ is an attention mask. For vanilla causal attention, $\boldsymbol{M}_{ij} = -\infty$ for $i < j$ and $\boldsymbol{M}_{ij} = 0$ for $i \ge j$. Finally, the output of final transformer block $\boldsymbol{H}^L$ is fed into an unembedding layer for prediction: $\boldsymbol{Y} = \text{LN}(\boldsymbol{H}^L)\boldsymbol{W}_{\text{cls}}$, where $\boldsymbol{W}_{\text{cls}} \in \mathbb{R}^{d \times |\mathbb{V}|}$.

**Positional embedding.** NoPE (Kazemnejad et al., 2024) considered no explicit positional embedding (PE) in LMs, where $\boldsymbol{P} = \boldsymbol{0}$. When using absolute PE (Vaswani et al., 2017), $\boldsymbol{P}$ is a periodic function of token positions. Devlin et al. (2019); Brown et al. (2020) adopted a learnable PE, which means $\boldsymbol{P}$ is a learnable embedding of token positions. The dot product between each query and key meets $\langle \boldsymbol{q}_i, \boldsymbol{k}_j \rangle = \boldsymbol{q}_i \boldsymbol{k}_j^\top$ when using the above three PEs. While for relative PE (Raffel et al., 2020), AL-iBi (Press et al., 2021), Rotary (Su et al., 2024), they have $\boldsymbol{P} = \boldsymbol{0}$. Instead, they modify the dot product $\langle \boldsymbol{q}_i, \boldsymbol{k}_j \rangle$. For relative PE and ALiBi, $\langle \boldsymbol{q}_i, \boldsymbol{k}_j \rangle = \boldsymbol{q}_i \boldsymbol{k}_j^\top + g(i - j)$, where $g(\cdot)$ is pre-defined function of the distance between two token positions. For Rotary, $\langle \boldsymbol{q}_i, \boldsymbol{k}_j \rangle = \boldsymbol{q}_i \boldsymbol{R}_{\Theta, j-i} \boldsymbol{k}_j^\top$, where $\boldsymbol{R}_{\Theta, (\cdot)}$ is a pre-defined rotation matrix. We include detailed formulations of the above PEs in Appendix B.

**Auto-regressive objective.** The pre-training objective of LMs is to maximize the likelihood of input data: $\theta^* = \arg\max_\theta \mathbb{E}_{\boldsymbol{X} \sim p_{\text{data}}} \big[ \sum_{t=1}^T \log p_\theta(\boldsymbol{x}_t | \boldsymbol{x}_{<t}) \big]$, where $p_{\text{data}}$ refers to the data distribution.

**Packing documents in pre-training.** Given a large corpus $\mathcal{D} = \{\boldsymbol{d}_1, \boldsymbol{d}_2, \cdots, \boldsymbol{d}_{|\mathcal{D}|}\}$, where each $\boldsymbol{d}_i$ represents a document containing a sequence of tokens. A packing strategy is adopted in the LM pre-training, as present in Figure 1(*Right*). All documents are concatenated and chunked into sequences with a context length of $C$. Each chunk could start with any token within one document or the BOS/EOS token. Then the empirical loss function for each chunk is $\mathcal{L} = \sum_{t=2}^C \log p_\theta(\boldsymbol{x}_t | \boldsymbol{x}_{<t})$. We note that $p_\theta(\boldsymbol{x}_1)$ is ignored since $\boldsymbol{y}_1 = f_\theta(\boldsymbol{x}_1)$ is the prediction for the next token $\boldsymbol{x}_2$.

**LM inference.** During the inference, a BOS token is fed into the model as the prefix for unconditional generation: $\boldsymbol{x}_t' \sim p_\theta(\boldsymbol{x}_t' | \boldsymbol{x}_{<t}', \boldsymbol{x}, \text{BOS})$, where $\boldsymbol{x}_t'$ is the $t$-th generated token, $\boldsymbol{x}$ is the optional prompt. If there are no BOS tokens in the pre-training, the EOS token is considered as the BOS.

**Attention sink.** Xiao et al. (2023b) revealed that LLMs allocate significant attention scores to specific token positions, e.g. the first token (not necessary to be a BOS token), resulting in "vertical" attention patterns. To represent this, we have the attention scores $\boldsymbol{A}_{i,1}^{l,h} \gg \text{mean}(\boldsymbol{A}_{i,j\neq1}^{l,h})$.

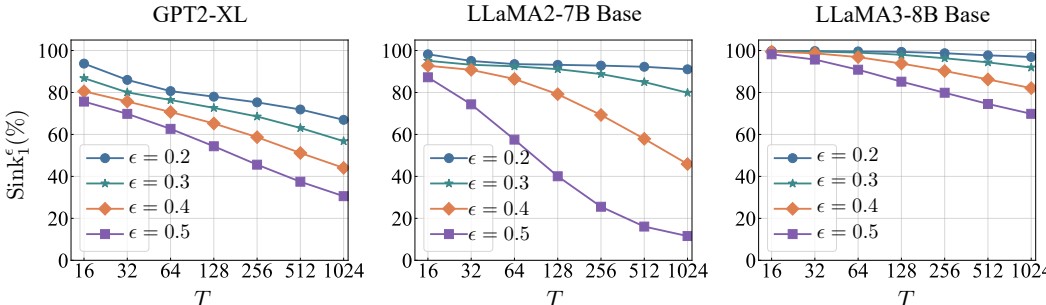

Figure 3: The metric $\text{Sink}_1^\epsilon$ (averaged on 100 sequences) tends to decrease with larger token lengths $T$. This tendency becomes more obvious with the more strict definition of attention sink (larger $\epsilon$).

# 3 PROPERTIES OF ATTENTION SINK

## 3.1 THE FIRST TOKEN ACTS AS BIASES

**Uniqueness of the first token.** It is noted that the calculation of hidden states for the first token has no involvement of self-attention $h_1^l = \text{FFN}(\text{LN}(o_1^l + h_1^{l-1})) + o_1^l + h_1^{l-1}$, where $o_1^l = \text{LN}(h_1^{l-1})[W^{l,1} \quad W^{l,2} \quad \cdots \quad W^{l,H}]W_O^l$. Therefore, $h_1^l$, and corresponding queries/keys/values $k_1^{l,h} = \text{LN}(h_1^{l-1})W_K^{l,h}$, $q_1^{l,h} = \text{LN}(h_1^{l-1})W_Q^{l,h}$, $v_1^{l,h} = \text{LN}(h_1^{l-1})W_V^{l,h}$ could be considered as the MLP output of input word embedding $x_1 W_E$. Using LLaMA3-8B Base (Dubey et al., 2024), we show that from certain transformer block, e.g., $l = 2$, the $\ell_2$-norm of $h_1^l$ is significantly larger than that of other tokens $h_{t \neq 1}$ in Figure 2(*Top*). This reproduces massive activations in Cancedda (2024); Sun et al. (2024). Despite the large $\ell_2$-norm of hidden states, we observe that the $\ell_2$-norm of keys and values of the first token is significantly smaller than that of other tokens in the same figure, which was also observed in Devoto et al. (2024); Guo et al. (2024b).

**QK angles contribute to attention sink.** In the $l$-th transformer block, we consider the keys and queries after adding PE (Rotary in LLaMA3-8B Base): $k_t^{l,h} = \text{LN}(h_t^{l-1})W_K^{l,h}R_{\Theta,-t}$, $q_t^{l,h} = \text{LN}(h_t^{l-1})W_Q^{l,h}R_{\Theta,-t}$, where LN is RMSNorm (Zhang & Sennrich, 2019): $\text{LN}(h) = \frac{h}{\text{RMS}(h)} \odot g$ and $\text{RMS}(h) = \sqrt{\frac{1}{d}\sum_{i=1}^{d}h_i^2}$. Here $g$ is a learnable gain parameter. Suppose that $h_1^{l-1}$ already has massive activations. Since $h_1^l$ has a massive magnitude in specific dimensions, the LN operation retains the magnitude in these dimensions and further reduces the magnitude in other dimensions, leading to that $q_1^{l,h}$, $k_1^{l,h}$, and $v_1^{l,h}$ are distributed on different manifolds, especially for $k_1^{l,h}$.

For the $t$-th query, we know that $q_t^{l,h}k_1^{l,h^\top}$ typically has much larger values than $q_t^{l,h}k_{j\neq 1}^{l,h^\top}$, as visualized in Figure 2(*Bottom*). We further show that due to the different manifold of $k_1^{l,h}$, the angles between $k_1^{l,h}$ and $q_t^{l,h}$ play an important role. Considering $q_t^{l,h}k_j^{l,h^\top} = \|q_t^{l,h}\| \cdot \|k_j^{l,h}\| \cdot \cos(q_t^{l,h}, k_j^{l,h})$, we visualize the cosine similarity between keys and values, and the product of $\ell_2$-norm between keys and values in Figure 2(*Bottom*). Although $\|q_t^{l,h}\| \cdot \|k_1^{l,h}\|$ is comparatively small, $\cos(q_t^{l,h}, k_1^{l,h})$ is significantly large, leading to attention sink. This explains why attention sink exists despite the small $\ell_2$-norm of keys of the first token. To conclude, the first token leverages its keys to act as biases, thus minimizing the angles between $k_1^{l,h}$ and $q_t^{l,h}$ and exhibiting attention sink.

## 3.2 MEASURING ATTENTION SINK

**Threshold-based metrics.** Xiao et al. (2023b) showcased the appearance of attention sink by visualizing attention logits/scores in different heads/blocks. This leads to the intractability of measuring attention sink quantitatively due to the large number of attention heads and blocks. Therefore, we first explore the metrics to measure the attention sink. Within each head, we compute the importance scores for the $k$-th token $\alpha_k^{l,h} = \frac{1}{T-k+1}\sum_{i=k}^{T}A_{i,k}^{l,h}$. We mainly focus on the first token $\alpha_1^{l,h}$. It is noted that $\frac{1}{T} \leq \alpha_1^{l,h} \leq 1$ since $A_{1,1}^{l,h} = 1$ and $0 \leq A_{i\neq 1,1}^{l,h} \leq 1$. Then we adopt a threshold-based metric, we consider a head has attention sink in the first token if $\alpha_1^{l,h} > \epsilon$. Considering that the whole model has $L$ blocks and each block has $H$ heads, we use the following metric to measure the attention sink of the whole LM: $\text{Sink}_k^\epsilon = \frac{1}{L}\sum_{l=1}^{L}\frac{1}{H}\sum_{h=1}^{H}\mathbb{I}(\alpha_k^{l,h} > \epsilon)$.

Table 1: (*Left*) Even with random sequence as input, there still exists an obvious attention sink. But with repeated tokens, the attention sink disappears for Mistral/LLaMA models. (*Right*) Chat models have comparable attention sink metrics with base models.

| LLM | $\text{Sink}_1^\epsilon(\%)$ | | | LLM | $\text{Sink}_1^\epsilon(\%)$ | |
|---|---|---|---|---|---|---|
| | natural | random | repeat | | Base | Chat |
| GPT2-XL | 77.00 | 70.29 | 62.28 | Mistral-7B | 97.49 | 88.34 |
| Mistral-7B | 97.49 | 75.21 | 0.00 | LLaMA2-7B | 92.47 | 92.88 |
| LLaMA2-7B Base | 92.47 | 90.13 | 0.00 | LLaMA2-13B | 91.69 | 90.94 |
| LLaMA3-8B Base | 99.02 | 91.23 | 0.00 | LLaMA3-8B | 99.02 | 98.85 |

Figure 4: (*Left*) Attention sink also emerges in small LMs. (*Middle*) Dynamics of train/valid loss and $\text{Sink}_1^\epsilon$ during LM pre-training under the default setup. Attention sink emerges after certain optimization steps. (*Right*) Training loss (solid lines)/attention sink (dashed lines) dynamics of LMs using different learning rates. We observe that with smaller learning rates, attention sink tends to emerge after more optimization steps and be less obvious.

**Selections of thresholds.** Typically, the selections of thresholds represent the strictness of quantifying attention sink. Generally, a larger $\epsilon$ represents a strict definition for attention sink. There is no principal way to find an optimal threshold and we only use this metric to quantify the emergence of attention sink empirically. Based on Figure 3, we prefer to select a threshold that is both strict in quantifying attention sink and less sensitive to the token length $T$. This gives us the selection of $\epsilon = 0.3$. For fair comparisons, we need to fix $T$ when computing the metric, e.g., $T = 64$.

### 3.3 ATTENTION SINK UNDER DIFFERENT INPUTS

**Different data domains.** We first explore the effects of input domains on attention sinks. The pile dataset (Gao et al., 2020), a regular dataset for LM pretraining, has 17 available data domains. As shown in Appendix C.2, input domains have negligible effects on our attention sink metric $\text{Sink}_1^\epsilon$.

**Beyond natural languages.** We also consider two ideal scenarios: (i) randomly sample $T$ tokens from the tokenizer vocabulary $\mathbb{V}$ to construct a sequence and (ii) randomly sample 1 token from the tokenizer $\mathbb{V}$ and repeat it $T$ times. As present in Table 1(*Left*), attention sink still exists when the inputs are random tokens instead of natural language. However, with repeated tokens, attention sink in Mistral (Jiang et al., 2023) and LLaMA models disappears. In Appendix C.1, we prove that for LMs with NoPE/relative PE/ALiBI/Rotary, if the first $T$ tokens are the same, their corresponding hidden states are the same. They all have massive activations, thus dispersing the attention sink. We also provide the closed form/upper bound for attention scores in these LMs through Propositions 1-4.

### 3.4 ATTENTION SINK UNDER DIFFERENT LMS

**Base vs. chat model.** Compared with base models, chat models are typically continually trained through instruction tuning (Ouyang et al., 2022). From Table 1(*Right*), instruction tuning has an insignificant impact on attention sink, which motivates us to focus on the LM pre-training.

**Model scale.** We evaluate the metric $\text{Sink}_1^\epsilon$ of LLaMA2 Base (Touvron et al., 2023), LLaMA3 Base (Dubey et al., 2024), Pythia (Biderman et al., 2023), GPT2 (Radford et al., 2019), OPT (Zhang et al., 2022) families. As visualized in Figure 4(*Left*), attention sink emerges in small LMs, even in Pythia-14M. Only in Pythia family, larger-sized LMs tend to have more obvious attention sink.

## 4 EFFECTS OF OPTIMIZATION ON ATTENTION SINK.

We pre-train a series of LLaMA models to conduct our experiments, based on the repos (Zhang et al., 2024a; Liu et al., 2024a). Due to the intractability of replicating LLaMA pre-training, we design small-sized models. Following Liu et al. (2024a), we set hidden dimension $d = 768$, block number $L = 10$, head number $H = 8$, intermediate size of FFN as 1536, resulting in approximately 60M parameters except for word embeddings and unembeddings. We keep the other design the same

Table 2: Larger weight decay ratios tend to induce more attention sink heads in LMs. But much larger values hurt the model performance and attention sink disappears.

| $\gamma$ | 0.0 | 0.001 | 0.01 | 0.1 | 0.5 | 1.0 | 2.0 | 5.0 |
|---|---|---|---|---|---|---|---|---|
| $\text{Sink}_1^\epsilon(\%)$ | 15.20 | 15.39 | 15.23 | 18.18 | 41.08 | 37.71 | 6.13 | 0.01 |
| valid loss | 3.72 | 3.72 | 3.72 | 3.73 | 3.80 | 3.90 | 4.23 | 5.24 |

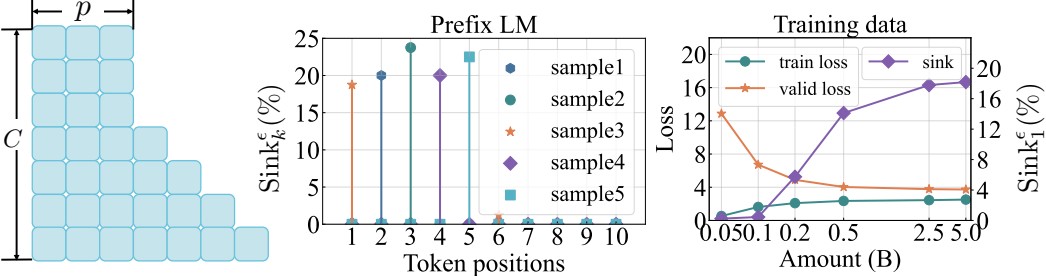

Figure 5: (*Left*) Attention pattern for prefix language modeling. (*Middle*) Attention sink does not only appear on the first token but among the prefix tokens for LMs with $p = 5$. (*Right*) With less training data, attention sink disappears. Meanwhile, trained LMs demonstrate overfitting behaviors.

as LLaMA2 models, including Rotary (Su et al., 2024), pre-norm structure, RMSNorm (Zhang & Sennrich, 2019) as LN, SwiGLU activation (Shazeer, 2020) in FFN, etc.

For data distribution, we sample 5B tokens from the Pile dataset (Gao et al., 2020). We set the context length to 2048 tokens, the batch size to 1M tokens, and the training step to 20k (including 100 steps for warm-up). We adopt a learning rate of 4e-4 with cosine scheduling. The optimizer is AdamW (Loshchilov & Hutter, 2017) with a weight decay ratio of 0.1. We use the Pile-CC validation loss (Gao et al., 2020; Liu et al., 2024a) to measure the model performance and sample 100 sequences with $T = 64$ (no BOS token) out of training data to measure the metric $\text{Sink}_k^\epsilon$ with $\epsilon = 0.3$.

**Optimization steps.** As visualized in Figure 4(*Middle*), under our default setup, attention sink emerges after certain optimization steps, e.g., between 1k and 2k steps. With the progression of pre-training, attention sink becomes more obvious.

**Learning rate.** With a smaller learning rate, it takes longer training steps to lower training loss, as present in Figure 4(*Right*). Meanwhile, the emergence of attention sink is also delayed. Besides, as shown in Table 9, we also find that a smaller learning rate results in LMs with less obvious attention sink, even if we compensate for more training steps. But further decreasing learning rate significantly affects the optimization and model performance, thus affecting the emergence of attention sink.

**Batch size.** In Table 10(*Left*), we find that only modifying batch size has no effects on attention sink.

> **Takeaways:** 1. Attention sink emerges after LMs are trained effectively. 2. Attention sink appears less obvious in LMs trained with small learning rates.

## 5 EFFECTS OF DATA DISTRIBUTION $p_{\text{DATA}}$ ON ATTENTION SINK

**Training data amount.** In the default setup, we consider 5B tokens. We wonder whether the attention sink emerges if we further constrain the data within a fixed compute budget. Therefore, we constrain the training data to 5B, 2.5B, 500M, 200M, 100M, and 50M. Meanwhile, we fix the batch size and optimization steps. As visualized in Figure 5(*Right*), with less training data, attention sink disappears. Further evidence in Figure 28 shows that this is not related to overfitting.

**Randomness in data distribution.** After packing documents into chunks, we re-sample the first token within the chunk $x_1 \sim \text{Uniform}(\mathbb{V})$. The trained LM has the metric $\text{Sink}_1^\epsilon = 27.03\%$, even larger than the default setup. This further validates the low semantic information of the sink token. We also consider $x_1, x_2 \sim \text{Uniform}(\mathbb{V})$, and we find attention sink shifts to the second token with $\text{Sink}_2^\epsilon = 14.08\%$ while the attention sink on the first token is much less obvious $\text{Sink}_1^\epsilon = 1.98\%$. But when only sample $x_2 \sim \text{Uniform}(\mathbb{V})$, the attention sink still always appears on the first token ($\text{Sink}_1^\epsilon = 20.99\%$). Additionally, we find with more random tokens during pre-training, attention sink tends to disappear.

**Fixing token in a specific position.** Xiao et al. (2023b) considered a learnable token in the first token position within each chunk, which can be considered as $x_1 \sim \mathbb{I}(x = x_{\text{fix}})$. We also consider fixing

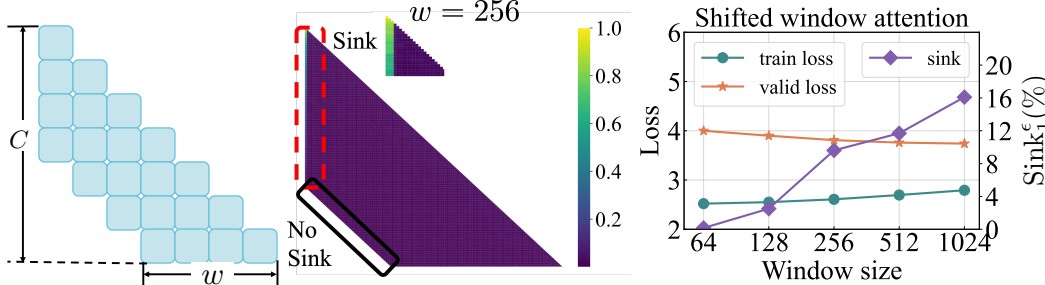

Figure 6: (*Left*) Shifted window attention pattern. (*Middle*) In LMs with window attention, attention sink appears on the first token, but not on the "first token" within each window. (*Right*) Attention sink tends to emerge when the window size is large enough.

the token $\boldsymbol{x}_{\text{fix}}$ in the second/third token position during pre-training. Consequently, the attention sink always appears in the fixed token instead of the first token, as shown in Table 10(*Right*).

> **Takeaways:** 1. Attention sink emerges after LMs are trained on sufficient training data. 2. Attention sink could be shifted to other positions rather than the first token if modifying $p_{\text{data}}$.

# 6 EFFECTS OF LOSS FUNCTION $\mathcal{L}$ ON ATTENTION SINK

**Weight decay.** The loss function becomes $\mathcal{L} = \sum_{t=2}^{C} \log p_\theta(\boldsymbol{x}_t | \boldsymbol{x}_{<t}) + \gamma \|\theta\|_2^2$ when introducing weight decay ratio $\gamma$. As indicated in Table 2, even $\gamma = 0$ in the loss function, attention sink still emerges in LMs. Then a larger $\gamma$ encourages more heads to have attention sink. But further increasing weight decay hurts the optimization, leading to less obvious or even no attention sink.

**Prefix language modeling.** Since the first token is not predicted in the auto-regressive loss function, it could be considered as the prefix token. Then the original auto-regressive loss can be generalized into the formula $\mathcal{L} = \sum_{t=p+1}^{C} \log p_\theta(\boldsymbol{x}_t | \boldsymbol{x}_{p+1:t-1}, \boldsymbol{x}_{1:p})$, with the prefix length $p = 1$. Motivated by Wang et al. (2022), we consider $p > 1$ and the casual mask visualized in Figure 5(*Left*). Although this design does not affect the emergence of attention sink, it shifts the sink position. In Figure 5(*Middle*), the attention sink only appears on one token. But it appears among these prefix tokens instead of on the first token only. Massive activations also appear on the corresponding sink token.

**Shifted window attention.** Motivated by the shifted window attention adopted in Mistral-7B, we further explore the effects of window size on attention sink. With shifted window attention, the loss function becomes $\mathcal{L} = \sum_{t=2}^{C} \log p_\theta(\boldsymbol{x}_t | \boldsymbol{x}_{t-w:t-1})$, where $w$ refers to the window size. As shown in Figure 6(*Left*) and (*Middle*), with shifted window attention, we find that if $t \leq w$, the $t$-th token can still "look at" the first token, and LMs still have attention sink on the first token. When $t > w$, the $t$-th token can only attend up to the $t - w + 1$-th token. Although this token is the "first token" for the $t$-th token, typically it has no attention sink. We have similar observations in Mistral-7B. Additionally, from Figure 6(*Right*), smaller window size prevents the emergence of attention sink.

> **Takeaways:** 1. Weight decay encourages the emergence of attention sink. 2. With prefix language modeling, attention sink appears among the prefix tokens rather than the first token only. 3. With shifted window attention, attention sink appears on the "absolute", not the "relative" first token. Smaller window size prevents the emergence of attention sink.

# 7 EFFECTS OF MODEL ARCHITECTURE $p_\theta$ ON ATTENTION SINK

In this section, we mainly explore the effects of positional embedding, pre-norm or post-norm structure, and attention design on the emergence of attention sink. In Appendix D, we also show that varying activation functions in the FFN, multi-head design do not affect the emergence of attention sink.

## 7.1 POSITIONAL EMBEDDING

Attention sink always appears on the first token, which motivates us to explore where such a position property is brought by positional embedding (PE). Therefore, we attempt to replace the original Rotary with other PEs, as shown in Table 3. We differentiate these PEs through the calculations of the hidden

Table 3: Positional embedding does not affect the emergence of attention sink.

| PE | $\boldsymbol{H}^0$ | $\langle \boldsymbol{q}_i, \boldsymbol{k}_j \rangle$ | $\text{Sink}_1^\epsilon(\%)$ | valid loss |
|---|---|---|---|---|
| NoPE | $\boldsymbol{X}\boldsymbol{W}_E$ | $\boldsymbol{q}_i\boldsymbol{k}_j^\top$ | 20.35 | 3.81 |
| Absolute PE | $\boldsymbol{X}\boldsymbol{W}_E + \boldsymbol{P}_{\text{abs}}$ | $\boldsymbol{q}_i\boldsymbol{k}_j^\top$ | 32.73 | 3.74 |
| Learnable PE | $\boldsymbol{X}\boldsymbol{W}_E + \boldsymbol{P}_{\text{learnable}}$ | $\boldsymbol{q}_i\boldsymbol{k}_j^\top$ | 33.13 | 3.79 |
| Relative PE | $\boldsymbol{X}\boldsymbol{W}_E$ | $\boldsymbol{q}_i\boldsymbol{k}_j^\top + g_{\text{relative}}(i - j)$ | 35.53 | 5.45 |
| ALiBi | $\boldsymbol{X}\boldsymbol{W}_E$ | $\boldsymbol{q}_i\boldsymbol{k}_j^\top + g_{\text{alibi}}(i - j)$ | 20.78 | 3.71 |
| Rotary | $\boldsymbol{X}\boldsymbol{W}_E$ | $\boldsymbol{q}_i\boldsymbol{R}_{\Theta, j-i}\boldsymbol{k}_j^\top$ | 18.18 | 3.73 |

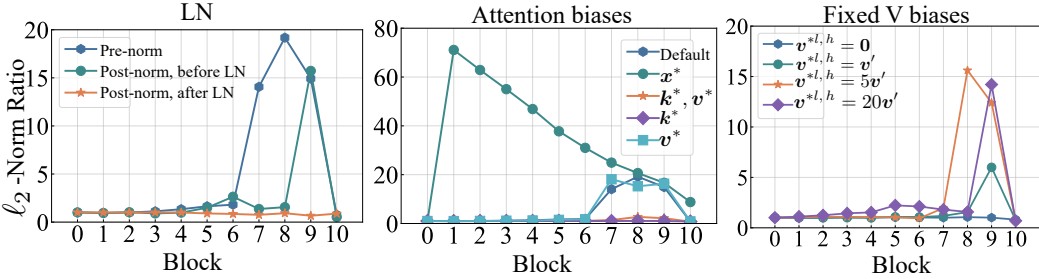

Figure 7: $\ell_2$-norm ratio of $\boldsymbol{h}_1^l$ and mean of $\boldsymbol{h}_{t\neq 0}^l$. (*Left*) Massive activations exist not in hidden states in post-norm LMs, but in the features before LN. (*Middle*) LMs with KV biases or K biases have no massive activations, while LMs with a learnable sink token or V biases have massive activations. (*Right*) Massive activations emerge when increasing the $\ell_2$-norm of fixed $\boldsymbol{v}^{*l, h}$ for the setup of K biases.

states $\boldsymbol{H}^0$ before the first transformer block and the dot product between queries and keys $\langle \boldsymbol{q}_i, \boldsymbol{k}_j \rangle$. The detailed formulations are delayed to Appendix B. From the same Table, we observe that only the model with relative PE is difficult to train while other models have comparable performance under our setup. Then we note that all these LMs, even the one without explicit PE (NoPE), have attention sink.

## 7.2 PRE-NORM AND POST-NORM STRUCTURE

Layer normalization (LN) (Ba et al., 2016; Zhang & Sennrich, 2019) regularizes the hidden states in LMs by re-centering and re-scaling, which may affect the massive activations. This motivates us to explore the effects of LN location on attention sink. In the pre-norm structure, as stated in Equation 2, hidden states from the earlier blocks could be retained by the later blocks through residual connections (He et al., 2016). Therefore, if massive activations appear in a specific block, they will likely be retained in the subsequent blocks. Within a post-norm structure, the hidden states will be normalized before being fed into the following blocks, as present in Figure 1(*Left*).

When replacing the pre-norm structure with the post-norm structure, $\text{Sink}_1^\epsilon$ becomes 13.54%. This indicates that the attention sink still exists in post-norm LMs. After further investigations, as visualized in Figure 7(*Left*), massive activations exist in the hidden states before the post LN instead of $\boldsymbol{h}_1^l$.

## 7.3 ATTENTION BIASES

**Learnable biases in attention.** In Section 3.1, we have shown that the first token acts as a bias: its key $\boldsymbol{k}_1^{l, h}$ is distributed in a different manifold and its value $\boldsymbol{v}_1^{l, h}$ has small $\ell_2$ norm. Xiao et al. (2023b) considered a learnable sink token in each chunk before the input tokens during LM pre-training. As this token is fixed in the first token, this could be considered as implicitly introducing biases $\boldsymbol{k}^{*l, h}$, $\boldsymbol{v}^{*l, h}$, $\boldsymbol{q}^{*l, h}$ in attention, as shown in the second row in Table 4. These biases are the MLP output of $\boldsymbol{x}^*\boldsymbol{W}_E$. Sun et al. (2024) directly introducing $\boldsymbol{k}^{*l, h}$ and $\boldsymbol{v}^{*l, h}$ as learnable parameters in attention (*KV biases*). They found that this design could alleviate the massive activations. Considering the important role of $\boldsymbol{k}^{*l, h}$ and small $\ell_2$-norm of $\boldsymbol{v}^{*l, h}$, we propose introducing only the key biases $\boldsymbol{k}^{*l, h}$ and fix value biases $\boldsymbol{v}^{*l, h} = \boldsymbol{0}$ (*K biases*). As a control, we also consider only adding value biases (*V biases*). The formulations of all these attention designs are shown in Table 4.

**LMs need key biases.** After evaluating the LMs with setups in Table 4, we first observe that these LMs have comparable model performance. Moreover, as long as there are key biases $\boldsymbol{k}^{*l, h}$, attention sink disappears on the first token but on the biases. From the setup of K biases, we reaffirm that the sink token acts as key biases, storing extra attention scores, which could be completely non-informative and not contribute to the value computation. It is worth mentioning that the introduced learnable sink token, KV biases, and V biases become part of model parameters in LMs. Removing them will lead to no attention sink in the first position, but a significant drop in model

Table 4: With comparable performance, LMs with sink token, KV biases, and K biases could shift attention sink from the first token to key biases' position. Value biases cannot affect attention sink.

| Attention in each head | $\text{Sink}_*^\epsilon(\%)$ | $\text{Sink}_1^\epsilon(\%)$ | valid loss |
|---|---|---|---|
| $\text{Softmax}\left(\frac{1}{\sqrt{d_h}} \boldsymbol{Q}^{l,h} \boldsymbol{K}^{l,h\top} + \boldsymbol{M}\right) \boldsymbol{V}^{l,h}$ | - | 18.18 | 3.73 |
| $\text{Softmax}\left(\frac{1}{\sqrt{d_h}} \begin{bmatrix} \boldsymbol{q}^{*l,h} \\ \boldsymbol{Q}^{l,h} \end{bmatrix} \begin{bmatrix} \boldsymbol{k}^{*l,h\top} & \boldsymbol{K}^{l,h\top} \end{bmatrix} + \boldsymbol{M}\right) \begin{bmatrix} \boldsymbol{v}^{*l,h} \\ \boldsymbol{V}^{l,h} \end{bmatrix}$ | 74.12 | 0.00 | 3.72 |
| $\text{Softmax}\left(\frac{1}{\sqrt{d_h}} \boldsymbol{Q}^{l,h} \begin{bmatrix} \boldsymbol{k}^{*l,h\top} & \boldsymbol{K}^{l,h\top} \end{bmatrix} + \boldsymbol{M}\right) \begin{bmatrix} \boldsymbol{v}^{*l,h} \\ \boldsymbol{V}^{l,h} \end{bmatrix}$ | 72.76 | 0.04 | 3.72 |
| $\text{Softmax}\left(\frac{1}{\sqrt{d_h}} \boldsymbol{Q}^{l,h} \begin{bmatrix} \boldsymbol{k}^{*l,h\top} & \boldsymbol{K}^{l,h\top} \end{bmatrix} + \boldsymbol{M}\right) \begin{bmatrix} \boldsymbol{0} \\ \boldsymbol{V}^{l,h} \end{bmatrix}$ | 73.34 | 0.00 | 3.72 |
| $\text{Softmax}\left(\frac{1}{\sqrt{d_h}} \boldsymbol{Q}^{l,h} \boldsymbol{K}^{l,h\top} + \boldsymbol{M}\right) \boldsymbol{V}^{l,h} + \boldsymbol{v}^{*l,h}$ | - | 17.53 | 3.73 |

Table 5: Larger $\ell_2$-norm of fixed $\boldsymbol{v}^{*l,h}$ results in LMs allocating more attention on $\boldsymbol{x}_1$ instead of $\boldsymbol{k}^{*l,h}$.

| $\boldsymbol{v}^{*l,h}$ | $\boldsymbol{0}$ | $\boldsymbol{v}'$ | $5\boldsymbol{v}'$ | $20\boldsymbol{v}'$ | $\boldsymbol{v}''$ | $5\boldsymbol{v}''$ | $20\boldsymbol{v}''$ |
|---|---|---|---|---|---|---|---|
| $\text{Sink}_*^\epsilon(\%)$ | 73.34 | 70.03 | 44.43 | 1.51 | 69.74 | 27.99 | 0.00 |
| $\text{Sink}_1^\epsilon(\%)$ | 0.00 | 0.06 | 3.71 | 25.88 | 2.15 | 5.93 | 11.21 |
| valid loss | 3.72 | 3.72 | 3.72 | 3.71 | 3.72 | 3.72 | 3.73 |

performance. In Figure 7(*Middle*), we also find that the LM with a learnable sink token has massive activations in $\boldsymbol{x}^*$. While LMs with KV biases and K biases have no massive activations.

**Beyond all zeros in V biases.** In the setup of K biases, we fix $\boldsymbol{v}^{*l,h} = \boldsymbol{0}$. We wonder whether the fixed values of $\boldsymbol{v}^{*l,h}$ could affect the attention sink. We consider, $\boldsymbol{v}^{*l,h} = m\boldsymbol{v}'$ or $\boldsymbol{v}^{*l,h} = m\boldsymbol{v}''$, where $m$ is the controllable $\ell_2$ norm and $\boldsymbol{v}' = [1, 0, 0, .., 0]$ and $\boldsymbol{v}'' = [1, 1, 1, .., 1]/\sqrt{d_h}$. As shown in Table 5, with larger $\ell_2$-norm of $\boldsymbol{v}^{*l,h}$, attention sink shifts from $\boldsymbol{k}^{*l,h}$ to the first token. Intuitively, it is difficult for LMs to remove the effects of $\boldsymbol{v}^{*l,h}$ with larger $\ell_2$-norm in model predictions. Then they opt to optimize the keys and values of the first token to save extra attention.

**Design space of biases.** In Table 12(*Right*), the LM with head-sharing KV biases tends to shift the sink from $\boldsymbol{k}^{*l,h}$ back to the first token. While the one with head-sharing K biases is less affected. In Table 13, even with small learnable dimensions for key biases, they can still absorb large attention.

### 7.4 ATTENTION OPERATION

**General formulation of attention.** In the last section, we realize that LMs need key biases to save extra attention. This motivates us to explore whether such a property is related to the dependence among attention scores due to the softmax operation. First, the attention output for the $i$-th token can be generalized as: $\boldsymbol{v}_i^\dagger = \left(\sum_{j'=1}^{i} \text{sim}(\varphi(\boldsymbol{q}_i), \varphi(\boldsymbol{k}_{j'}))\right)^{-1} \sum_{j=1}^{i} \text{sim}(\varphi(\boldsymbol{q}_i), \varphi(\boldsymbol{k}_j))\boldsymbol{v}_j = \boldsymbol{Z}_i^{-1} \sum_{j=1}^{i} \text{sim}(\varphi(\boldsymbol{q}_i), \varphi(\boldsymbol{k}_j))\boldsymbol{v}_j$, where we omit the PE, and block/head indexes for simplicity. $\boldsymbol{Z}_i$ is a normalization term and $\varphi(\cdot)$ is a kernel function. Normally, $\boldsymbol{Z}_i = \sum_{j'=1}^{i} \text{sim}(\varphi(\boldsymbol{q}_i), \varphi(\boldsymbol{k}_j))$. For softmax attention, $\varphi(\cdot)$ is an identity kernel and $\text{sim}(\boldsymbol{q}_i, \boldsymbol{k}_j) = \exp(\boldsymbol{q}_i \boldsymbol{k}_j^\top / \sqrt{d_h})$.

**Normalization.** In Table 14(*Left*) and Figure 29, we show that modifying normalization may result in less obvious attention sink but does not stop its emergence. So we consider removing the normalization. Since the exponential function in softmax tends to explode without normalization, we replace it with sigmoid or elu plus one. When evaluating the attention sink, we compute the *proxy* attention scores by using the term $\boldsymbol{Z}_i = \sum_{j'=1}^{i} \text{sim}(\varphi(\boldsymbol{q}_i), \varphi(\boldsymbol{k}_{j'}))$ for attention sink metric $\text{Sink}_1^\epsilon$. As shown in Table 6, without normalization, LMs still have comparable validation loss but no attention sink. With normalization, attention sink also emerges in LMs with sigmoid attention.

**Kernel functions.** Motivated by linear attention (Katharopoulos et al., 2020), we consider different kernel functions $\varphi(\cdot)$, including elu plus one, identity, and MLP. It is noted that $\text{sim}(\varphi(\boldsymbol{q}_i), \varphi(\boldsymbol{k}_{j'}))$ could be minus for identity and MLP kernels. This brings intrinsic difficulty for normalization during the training and calculation of $\text{Sink}_1^\epsilon$. For normalization in the training, we consider $\boldsymbol{Z}_i = \max\left(\left|\sum_{j'=1}^{i} \text{sim}(\varphi(\boldsymbol{q}_i), \varphi(\boldsymbol{k}_{j'}))\right|, 1\right)$. When computing $\text{Sink}_1^\epsilon$, we consider the proxy attention scores $|\text{sim}(\varphi(\boldsymbol{q}_i), \varphi(\boldsymbol{k}_j))| / \sum_{j'=1}^{i} |\text{sim}(\varphi(\boldsymbol{q}_i), \varphi(\boldsymbol{k}_{j'}))|$. From Table 6 (a full version in Table 15), we find that the LM with MLP kernel have no attention sink with or without normalization.

Table 6: Normalization and selections of kernels in attention significantly affect the emergence of the attention sink. We use "*" to mark that the metric $\text{Sink}_1^\epsilon$ is computed by proxy attention scores.

| $\text{sim}(\varphi(\boldsymbol{q}_i), \varphi(\boldsymbol{k}_j))$ | $\boldsymbol{Z}_i$ | $\text{Sink}_1^\epsilon$(%) | valid loss |
|---|---|---|---|
| $\exp(\frac{\boldsymbol{q}_i\boldsymbol{k}_j^\top}{\sqrt{d_h}})$ | $\sum_{j'=1}^{i} \exp(\frac{\boldsymbol{q}_i\boldsymbol{k}_{j'}^\top}{\sqrt{d_h}})$ | 18.18 | 3.73 |
| $\text{sigmoid}(\frac{\boldsymbol{q}_i\boldsymbol{k}_j^\top}{\sqrt{d_h}})$ | 1 | 0.44* | 3.70 |
| $\text{sigmoid}(\frac{\boldsymbol{q}_i\boldsymbol{k}_j^\top}{\sqrt{d_h}})$ | $\sum_{j'=1}^{i} \text{sigmoid}(\frac{\boldsymbol{q}_i\boldsymbol{k}_{j'}^\top}{\sqrt{d_h}})$ | 30.24 | 3.74 |
| $\text{elu}(\frac{\boldsymbol{q}_i\boldsymbol{k}_j^\top}{\sqrt{d_h}})+1$ | 1 | 0.80* | 3.69 |
| $\frac{(\text{elu}(\boldsymbol{q}_i)+1)(\text{elu}(\boldsymbol{k}_j)+1)^\top}{\sqrt{d_h}}$ | $\sum_{j'=1}^{i} \frac{(\text{elu}(\boldsymbol{q}_i)+1)(\text{elu}(\boldsymbol{k}_{j'})+1)^\top}{\sqrt{d_h}}$ | 53.65* | 4.19 |
| $\frac{\boldsymbol{q}_i\boldsymbol{k}_j^\top}{\sqrt{d_h}}$ | 1 | 0.00* | 3.99 |
| $\frac{\text{mlp}(\boldsymbol{q}_i)\text{mlp}(\boldsymbol{k}_j)^\top}{\sqrt{d_h}}$ | $\max\left(\left|\sum_{j'=1}^{i} \frac{\text{mlp}(\boldsymbol{q}_i)\text{mlp}(\boldsymbol{k}_{j'})^\top}{\sqrt{d_h}}\right|, 1\right)$ | 0.19* | 3.85 |
| $\frac{\text{mlp}(\boldsymbol{q}_i)\text{mlp}(\boldsymbol{k}_j)^\top}{\sqrt{d_h}}$ | 1 | 0.74* | 3.91 |

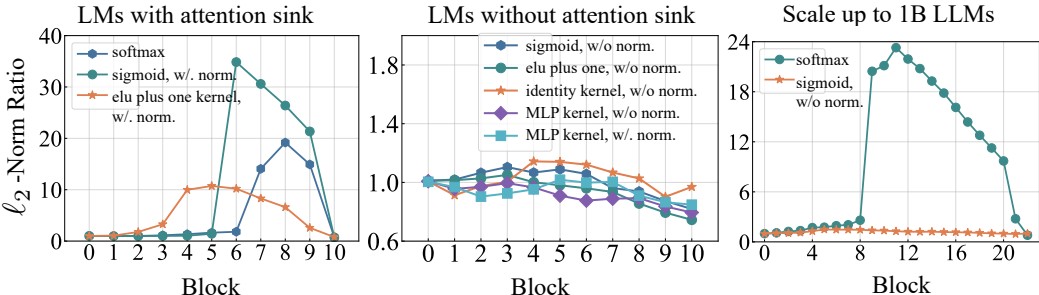

Figure 8: We visualize the $\ell_2$-norm ratio of $\boldsymbol{h}_1^l$ and mean of $\boldsymbol{h}_{t\neq0}^l$. (*Left*) Massive activations exist in LMs with attention scores that are non-negative and added up to one. (*Middle*) Massive activations do not exist in LMs with independent attention scores. (*Right*) When scaling the model size to 1B, LLMs with sigmoid attention (no normalization) still have no massive activations.

**Inner dependence on attention scores.** We note that the LMs with no attention sink typically relax tokens' inner dependence on attention scores. Their attention scores during pre-training could be negative or not add up to one. This indicated that attention sink (at least partially) stems from such inner dependence. Besides the attention metric computed by proxy attention scores, we also observe that the above LMs also have no massive activations, as shown in Figure 8(*Middle*).

**Scale up to 1B parameters.** We compare model behaviors of 1B LMs with softmax attention and sigmoid attention (without normalization). Specifically, the latter achieves a validation loss of 3.10, slightly larger than the 3.07 achieved by the former. However, the attention sink metric significantly drops from 45.11% to near zero: $\text{Sink}_1^\epsilon = 2.46\%$ using the proxy attention scores. Meanwhile, as present in Figure 8(*Right*), LLMs with sigmoid attention have no massive activations. Furthermore, they have no issues of training stability during continued supervised fine-tuning in Appendix E.

> **Takeaways:** 1. Positional embedding, FFN design, LN location, and multi-head design do not affect the emergence of attention sink. 2. Attention sink acts more like key biases, storing extra attention and meanwhile not contributing to the value computation. 3. When relaxing tokens' inner dependence on attention scores, attention sink does not emerge in LMs.

# 8 FUTURE WORK

This work focuses on the sink token in the first position. Sun et al. (2024); Yu et al. (2024) showed that attention sink can also appear on certain word tokens, e.g., period and newline tokens. However, these sink words may vary in different LMs and typically have no fixed positions. In the future, we will extend the research scope to explore how these sink tokens are related to the pre-training. Additionally, it remains unclear whether attention sink benefits LM downstream performance.

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

# A RELATED WORK

## A.1 ATTENTION SINK PHENOMENON

Although the term *attention sink* was introduced by Xiao et al. (2023b), similar observations have been reported in prior studies. Among them, Zhai et al. (2023) demonstrated that attention weights in Transformer models (Vaswani et al., 2017) collapse into one-hot vectors, a phenomenon termed *attention entropy collapse*. This issue has been identified as *attention logit growth* by Wortsman et al. (2023); Dehghani et al. (2023). Furthermore, Bondarenko et al. (2023) observed the emergence of strong outliers in Transformer models, linked to attention heads that learn not to update residuals. Although not explicitly using the term attention sink, Han et al. (2024) highlighted the significance of the first few tokens in language models (LMs), noting their distinct representational space. In vision transformers (ViTs), Darcet et al. (2023) identified high-norm outlier tokens as artifacts in attention maps. Overall, the attention sink phenomenon is prevalent across Transformer models, including ViTs, encoder-only LMs, such as BERT (Devlin et al., 2019), and auto-regressive LMs. Notably, in auto-regressive LMs, attention sink consistently occurs on the first token in addition to tokens with limited semantic information in the middle context (Sun et al., 2024; Yu et al., 2024).

## A.2 ATTENTION SINK AND ACTIVATION OUTLIERS

Cancedda (2024) found that early FFNs in LLaMA2 blast off the large norm of hidden states for the first token, leading to attention sink in later layers. This is referred to as *massive activations* (very few activations exhibit disproportionately large values) in Sun et al. (2024) and *token-wise activation outlier* in outlier literature (Lin et al., 2024). Similar observations were made by Bondarenko et al. (2023). In contrast, *channel-wise activation outlier* (Dettmers et al., 2022; Xiao et al., 2023a) refer to that outliers appear in specific channels across all token positions. A concurrent study (Kaul et al., 2024) found that these two types of outliers are distinct and require different mitigation strategies. Despite the high magnitude in hidden states, Bondarenko et al. (2023); Guo et al. (2024b) observed the values associated with sink tokens are typically smaller than those of other tokens.

## A.3 UNDERSTANDING AND MITIGATE ATTENTION SINK

To understand the attention sink phenomenon in general Transformer models, Bondarenko et al. (2023) hypothesized that the interplay among softmax, residual connections, and LayerNorm encourages models to learn not to update residuals. In auto-regressive LMs, Sun et al. (2024) attributed attention sink at the first position to implicit biases in keys and values. A concurrent work by Guo et al. (2024a) provided empirical and theoretical analyses of attention sink in very simple Transformer LMs on the Bigram-Backcopy (BB) task, identifying a mutual reinforcement mechanism as the underlying cause. Their findings on the BB task indicated that (i) replacing softmax with ReLU removes attention sink; (ii) switching the optimizer Adam to SGD eliminates massive activations but attention sink remains.

Besides the above literature, Xiao et al. (2023b); Kaul et al. (2024) found that replacing softmax with softmax-off-by-one (Miller, 2023) alleviates attention sink at the first position, though Xiao et al. (2023b) noted its potential emergence at other initial token positions. Yin et al. (2024) proposed refining the casual mask with pseudo-attention values on the "future tokens", which could reduce attention sink. Though not explicitly targeting attention sink, methods such as $\sigma$Reparam (Zhai et al., 2023) and qk-norm (Dehghani et al., 2023) were introduced to address attention entropy collapse. To mitigate activation outliers, He et al. (2024) developed an outlier-protected block and also highlighted the role of optimization in mitigating the outliers. Besides, Bondarenko et al. (2023) proposed clipped softmax and gated attention, while Kaul et al. (2024) introduced OrthoAdam to reduce activation outliers.

## A.4 APPLICATIONS OF ATTENTION SINK

Sink tokens in LMs are functionally important and absorb significant attention. This enables computational savings by prioritizing initial and recent tokens over middle-context tokens in attention calculations. Token-wise activation outliers pose challenges for quantization, necessitating specialized handling to facilitate quantization processes. These insights have motivated various applications, including streaming/long context generation (Xiao et al., 2023b; Han et al., 2024; Yang et al., 2024), KV cache optimization (Ge et al., 2023; Wan et al., 2024; Wu & Tu, 2024), efficient inference (Zhang et al., 2024b; Chen et al., 2024), model quantization (Liu et al., 2024b; Huang et al., 2024), and others.

## B  DETAILED FORMULATIONS OF POSITIONAL EMBEDDING

In this section, we provide detailed formulations of positional embedding (PE) in LMs. PEs could be classified into two categories. NoPE, absolute positional embedding, and learnable positional embedding belong to the same category since they are added to the initial hidden states: $\boldsymbol{H}^0 = \boldsymbol{X}\boldsymbol{W}_E + \boldsymbol{P}$. Here $\boldsymbol{P} = \{\boldsymbol{p}_1, \boldsymbol{p}_2, \cdots, \boldsymbol{p}_T\} \in \mathbb{R}^{T \times d}$. Meanwhile, the dot product between each query and key is computed as $\langle \boldsymbol{q}_i, \boldsymbol{k}_j \rangle = \boldsymbol{q}_i \boldsymbol{k}_j^\top$.

**NoPE.** NoPE (Kazemnejad et al., 2024) refers to no positional embedding. Therefore, $\boldsymbol{P} = \boldsymbol{0}$.

**Absolute PE.** Each position vector $\boldsymbol{p}_t$ in absolute positional embedding is a periodic function of token position $t$ following Vaswani et al. (2017):

$$\boldsymbol{p}_t = \begin{bmatrix} \sin(\omega_1 t) & \cos(\omega_1 t) & \sin(\omega_2 t) & \cos(\omega_2 t) & \cdots & \sin(\omega_{d/2} t) & \cos(\omega_{d/2} t) \end{bmatrix}, \tag{5}$$

where $\omega_i = 1/10000^{2(i-1)/d}$.

**Learnable PE.** Each position vector $\boldsymbol{p}_t$ in learnable positional embeddings is a learnable parameter.

Relative positional embedding, ALibi, and rotary belong to another category since they consider the relative distance among tokens. Therefore, the initial hidden states are $\boldsymbol{H}^0 = \boldsymbol{X}\boldsymbol{W}_E$ but how to compute the dot product between each query and key is modified.

**Relative PE.** The relative positional embeddings are adopted in T5 models (Raffel et al., 2020). A bias term is adopted for the dot product: $\langle \boldsymbol{q}_i, \boldsymbol{k}_j \rangle = \boldsymbol{q}_i \boldsymbol{k}_j^\top + g(i-j)$, where the definition for distance function $g(\cdot)$ is:

$$g(i-j) = \begin{cases} i-j & \text{if } i-j < \mathcal{B}/2 \\ \frac{\mathcal{B}}{2} + \lfloor \frac{\log(\frac{i-j}{\mathcal{B}/2})}{\log(\frac{\mathcal{D}}{\mathcal{B}/2})} \rfloor \times \frac{\mathcal{B}}{2} & \text{if } \mathcal{B}/2 \leq i-j < \mathcal{D} \\ \mathcal{B} - 1 & \text{if } i-j \geq \mathcal{D} \end{cases} \tag{6}$$

Here $\mathcal{B}$ and $\mathcal{D}$ refer to the number of buckets and maximum distance, respectively. In T5 models, $\mathcal{B} = 32$ and $\mathcal{D} = 128$.

**ALiBi.** Similarly, ALiBi (Press et al., 2021) also adds a bias term to the dot product: $\langle \boldsymbol{q}_i, \boldsymbol{k}_j \rangle = \boldsymbol{q}_i \boldsymbol{k}_j^\top + g(i-j)$, where $g(i-j) = -(i-j) \cdot m$. $m$ is a head-specific slope fixed:

$$m = 2^{-h \cdot 2^{-\log_2 H + 3}}, \tag{7}$$

where $1 \leq h \leq H$ is the head index and $H$ is the number of heads in the multi-head self-attention (MHSA). This slope $m$ is a geometric sequence. For instance, when $H = 8$, the sequence is $\frac{1}{2^1}, \frac{1}{2^2}, \cdots, \frac{1}{2^8}$. When $H = 16$, the sequence is $\frac{1}{2^{0.5}}, \frac{1}{2^1}, \frac{1}{2^{1.5}}, \cdots, \frac{1}{2^8}$.

**Rotary.** Rotary (Su et al., 2024) is the most adopted position encoding approach in the LLM community. It projects queries and keys into another space through rotations:

$$\langle \boldsymbol{q}_i, \boldsymbol{k}_j \rangle = (\boldsymbol{q}_i \boldsymbol{R}_{\Theta, -i})(\boldsymbol{k}_j \boldsymbol{R}_{\Theta, -j})^\top = \boldsymbol{q}_i \boldsymbol{R}_{\Theta, -i} \boldsymbol{R}_{\Theta, j} \boldsymbol{k}_j^\top = \boldsymbol{q}_i \boldsymbol{R}_{\Theta, j-i} \boldsymbol{k}_j^\top, \tag{8}$$

where $\boldsymbol{R}_{\Theta, (\cdot)}$ is a pre-defined rotation matrix:

$$\boldsymbol{R}_{\Theta, m} = \begin{pmatrix} \cos m\omega_1 & -\sin m\omega_1 & 0 & 0 & \cdots & 0 & 0 \\ \sin m\omega_1 & \cos m\omega_1 & 0 & 0 & \cdots & 0 & 0 \\ 0 & 0 & \cos m\omega_2 & -\sin m\omega_2 & \cdots & 0 & 0 \\ 0 & 0 & \sin m\omega_2 & \cos m\omega_2 & \cdots & 0 & 0 \\ \vdots & \vdots & \vdots & \vdots & \ddots & \vdots & \vdots \\ 0 & 0 & 0 & 0 & \cdots & \cos m\omega_{d_h/2} & -\sin m\omega_{d_h/2} \\ 0 & 0 & 0 & 0 & \cdots & \sin m\omega_{d_h/2} & \cos m\omega_{d_h/2} \end{pmatrix}. \tag{9}$$

Here $\omega_i = 1/10000^{2(i-1)/d_h}$ and $d_h = d/H$ is the hidden dimension in each head. From the above definition, it is noted that the rotation matrix $\boldsymbol{R}_{\Theta, m}^{d_h}$ satisfies $\boldsymbol{R}_{\Theta, m}^\top = \boldsymbol{R}_{\Theta, -m}$ and $\boldsymbol{R}_{\Theta, i} \boldsymbol{R}_{\Theta, j} = \boldsymbol{R}_{\Theta, i+j}$.

## C  ATTENTION SINK IN OPEN-SOURCED LMs

### C.1  HOW POSITIONAL EMBEDDING RELATES TO ATTENTION SINK

In Section 3.3, we have shown that Mistral-7B, LLaMA2-7B Base, and LLaMA3-8B Base, which adopt rotary as PE, have no attention sink for repeated token sequences. While GPT2, which adopts learnable PE, has the attention sink in such a scenario. To further understand this, we explore how positional embedding plays a role through a theoretical perspective.

**Activations.** Suppose the repeated sequence is $\boldsymbol{X} = \{\boldsymbol{x}_1, \boldsymbol{x}_2, \ldots, \boldsymbol{x}_T\}$ and each $\boldsymbol{x}_t = \boldsymbol{x}$. For LMs with NoPE/relative PE/ALiBi/Rotary, we have the initial hidden states $\boldsymbol{h}_t^0 = \boldsymbol{x}\boldsymbol{W}_E$, which are the same among all the token positions since $\boldsymbol{P} = \boldsymbol{0}$. Then for the first transformer block $l = 1$, we know that $\boldsymbol{k}_t^{1,h} = \text{LN}(\boldsymbol{h}_t^0)\boldsymbol{W}_K^{1,h} = \text{LN}(\boldsymbol{x}\boldsymbol{W}_E)\boldsymbol{W}_K^{1,h}$, $\boldsymbol{q}_t^{1,h} = \text{LN}(\boldsymbol{h}_t^0)\boldsymbol{W}_Q^{1,h} = \text{LN}(\boldsymbol{x}\boldsymbol{W}_E)\boldsymbol{W}_Q^{1,h}$, $\boldsymbol{v}_t^{1,h} = \text{LN}(\boldsymbol{h}_t^0)\boldsymbol{W}_V^{1,h} = \text{LN}(\boldsymbol{x}\boldsymbol{W}_E)\boldsymbol{W}_V^{1,h}$. Then all tokens have the same $\boldsymbol{k}_t^{1,h}$, $\boldsymbol{q}_t^{1,h}$, and $\boldsymbol{v}_t^{1,h}$. Then the attention output is

$$\boldsymbol{v}_t^{\dagger 1,h} = \sum_{i=1}^t \boldsymbol{A}_{ti}^{1,h} \boldsymbol{v}_i^{1,h} = \boldsymbol{v}_t^{1,h} \tag{10}$$

Then $\boldsymbol{o}_t^1 = \text{Concat}_{h=1}^H(\boldsymbol{v}_t^{1,h})\boldsymbol{W}_O^1$, we have the hidden states after the first block:

$$\boldsymbol{h}_t^1 = \text{FFN}(\text{LN}(\boldsymbol{o}_t^1 + \boldsymbol{h}_t^0)) + \boldsymbol{o}_t^1 + \boldsymbol{h}_t^0 \tag{11}$$

$$= \text{FFN}(\text{LN}(\text{Concat}_{h=1}^H(\boldsymbol{v}_t^{1,h})\boldsymbol{W}_O^1 + \boldsymbol{h}_t^0)) + \text{Concat}_{h=1}^H(\boldsymbol{v}_t^{1,h})\boldsymbol{W}_O^1 + \boldsymbol{h}_t^0 \tag{12}$$

$$= \text{FFN}(\text{LN}(\text{Concat}_{h=1}^H(\text{LN}(\boldsymbol{h}_t^0)\boldsymbol{W}_V^{1,h})\boldsymbol{W}_O^1 + \boldsymbol{h}_t^0)) \tag{13}$$

$$+ \text{Concat}_{h=1}^H(\text{LN}(\boldsymbol{h}_t^0)\boldsymbol{W}_V^{1,h})\boldsymbol{W}_O^1 + \boldsymbol{h}_t^0. \tag{14}$$

Since $\boldsymbol{h}_1^0 = \boldsymbol{h}_2^0 = \cdots = \boldsymbol{h}_T^0$, we have $\boldsymbol{h}_1^1 = \boldsymbol{h}_2^1 = \cdots = \boldsymbol{h}_T^1$ based on the above equation. Using this induction, we could prove that

$$\boldsymbol{h}_t^l = \text{FFN}(\text{LN}(\text{Concat}_{h=1}^H(\text{LN}(\boldsymbol{h}_t^{l-1})\boldsymbol{W}_V^{l,h})\boldsymbol{W}_O^l + \boldsymbol{h}_t^{l-1})) \tag{15}$$

$$+ \text{Concat}_{h=1}^H(\text{LN}(\boldsymbol{h}_t^{l-1})\boldsymbol{W}_V^{l,h})\boldsymbol{W}_O^l + \boldsymbol{h}_t^{l-1}, \ \forall \ 1 \leq l \leq L. \tag{16}$$

$$\boldsymbol{h}_1^l = \boldsymbol{h}_2^l = \cdots = \boldsymbol{h}_T^l, \ \forall \ 0 \leq l \leq L. \tag{17}$$

Typically, the hidden states of the first token $\boldsymbol{h}_1^l$ in specific blocks have massive activations. Due to the above equality, all the repeated tokens have massive activations. Furthermore, we could derive the closed form or upper bounds for attention scores under the repeated token sequence.

**Proposition 1.** *For LMs with NoPE, the attention scores for $t$ repeated tokens are $t^{-1}$ uniformly, i.e., there is no attention sink.*

*Proof.* We have that

$$\boldsymbol{A}_{ti}^{l,h} = \frac{e^{\langle \boldsymbol{q}_t^{l,h}, \boldsymbol{k}_i^{l,h}\rangle}}{\sum_{j=1}^t e^{\langle \boldsymbol{q}_t^{l,h}, \boldsymbol{k}_j^{l,h}\rangle}} = \frac{e^{\boldsymbol{q}_t^{l,h}\boldsymbol{k}_i^{l,h\top}}}{\sum_{j=1}^t e^{\boldsymbol{q}_t^{l,h}\boldsymbol{k}_j^{l,h\top}}} = \frac{e^{\boldsymbol{q}^{l,h}\boldsymbol{k}^{l,h\top}}}{te^{\boldsymbol{q}^{l,h}\boldsymbol{k}^{l,h\top}}} = \frac{1}{t}. \tag{18}$$

Therefore, the attention scores follow a uniform distribution over all previous tokens. □

**Proposition 2.** *For LMs with relative PE, there is no attention sink for $t$ repeated tokens.*

*Proof.* For LMs with relative PE, the dot product between each query and key is

$$\langle \boldsymbol{q}_t^{l,h}, \boldsymbol{k}_i^{l,h}\rangle = \boldsymbol{q}_t^{l,h}\boldsymbol{k}_i^{l,h\top} + g_{\text{rel}}(t-i) = \boldsymbol{q}^{l,h}\boldsymbol{k}^{l,h\top} + g_{\text{rel}}(t-i), \tag{19}$$

then we have the attention scores

$$\boldsymbol{A}_{t,i}^{l,h} = \frac{e^{\langle \boldsymbol{q}_t^{l,h}, \boldsymbol{k}_i^{l,h}\rangle}}{\sum_{j=1}^t e^{\langle \boldsymbol{q}_t^{l,h}, \boldsymbol{k}_j^{l,h}\rangle}} = \frac{e^{\boldsymbol{q}^{l,h}\boldsymbol{k}^{l,h\top} + g_{\text{rel}}(t-i)}}{\sum_{j=1}^t e^{\boldsymbol{q}^{l,h}\boldsymbol{k}^{l,h\top} + g_{\text{rel}}(t-j)}} = \frac{e^{g_{\text{rel}}(t-i)}}{\sum_{j=1}^t e^{g_{\text{rel}}(t-j)}}. \tag{20}$$

Considering $g_{\text{rel}}(t-i)$ is a monotonic non-increasing function of $t-i$ and $g_{\text{rel}}(t-i) = \mathcal{B} - 1$ when $t - i > \mathcal{D}$, then $\boldsymbol{A}_{t,1}^{l,h} = \boldsymbol{A}_{t,1}^{l,h} = \cdots = \boldsymbol{A}_{t,t-\mathcal{D}}^{l,h}$ are the largest values. Therefore, there is no attention sink on the first token. $\qquad\square$

**Proposition 3.** *For LMs with ALiBi, there is no attention sink for $t$ repeated tokens.*

*Proof.* For LMs with ALiBi, similar to relative PE, the dot product between each query and key is

$$\langle \boldsymbol{q}_t^{l,h}, \boldsymbol{k}_i^{l,h} \rangle = \boldsymbol{q}_t^{l,h} \boldsymbol{k}_i^{l,h^\top} + g_{\text{alibi}}^h(t-i) = \boldsymbol{q}^{l,h} \boldsymbol{k}^{l,h^\top} + g_{\text{alibi}}^h(t-i), \tag{21}$$

then we have the attention scores

$$\boldsymbol{A}_{t,i}^{l,h} = \frac{e^{\langle \boldsymbol{q}_t^{l,h}, \boldsymbol{k}_i^{l,h} \rangle}}{\sum_{j=1}^t e^{\langle \boldsymbol{q}_t^{l,h}, \boldsymbol{k}_j^{l,h} \rangle}} = \frac{e^{\boldsymbol{q}^{l,h} \boldsymbol{k}^{l,h^\top} + g_{\text{alibi}}^h(t-i)}}{\sum_{j=1}^t e^{\boldsymbol{q}^{l,h} \boldsymbol{k}^{l,h^\top} + g_{\text{alibi}}^h(t-j)}} = \frac{e^{g_{\text{alibi}}^h(t-i)}}{\sum_{j=1}^t e^{g_{\text{alibi}}^h(t-j)}}. \tag{22}$$

Here $g_{\text{alibi}}^h(t-i)$ is monotonic decreasing function of $t-i$, so there is no attention sink on the first token. $\qquad\square$

**Proposition 4.** *For LMs with Rotary, there is no attention sink for $t$ repeated tokens when $t$ is large if $\|\boldsymbol{q}^{l,h}\| \cdot \|\boldsymbol{k}^{l,h}\| \le \xi$ for a constant $\xi$.*

*Proof.* For LMs with Rotary, the dot product between each query and key is

$$\langle \boldsymbol{q}_t^{l,h}, \boldsymbol{k}_i^{l,h} \rangle = \boldsymbol{q}_t^{l,h} \boldsymbol{R}_{\Theta, i-t} \boldsymbol{k}_i^{l,h^\top} \tag{23}$$

$$= \boldsymbol{q}^{l,h} \boldsymbol{R}_{\Theta, i-t} \boldsymbol{k}^{l,h^\top} \tag{24}$$

$$= \left\| \boldsymbol{q}^{l,h} \right\| \left\| \boldsymbol{k}^{l,h} \boldsymbol{R}_{\Theta, t-i} \right\| \cos \left( \frac{\boldsymbol{q}^{l,h} \boldsymbol{R}_{\Theta, i-t} \boldsymbol{k}^{l,h^\top}}{\left\| \boldsymbol{q}^{l,h} \right\| \left\| \boldsymbol{k}^{l,h} \boldsymbol{R}_{\Theta, t-i} \right\|} \right) \tag{25}$$

$$= \left\| \boldsymbol{q}^{l,h} \right\| \left\| \boldsymbol{k}^{l,h} \right\| \cos(\beta_{t-i}), \tag{26}$$

where $\beta_{j-t}$ is the angle between the rotated query and the rotated key. Then the attention scores are

$$\boldsymbol{A}_{t,i}^{l,h} = \frac{e^{\langle \boldsymbol{q}_t^{l,h}, \boldsymbol{k}_i^{l,h} \rangle}}{\sum_{j=1}^t e^{\langle \boldsymbol{q}_t^{l,h}, \boldsymbol{k}_j^{l,h} \rangle}} = \frac{e^{\boldsymbol{q}^{l,h} \boldsymbol{R}_{\Theta, j-i} \boldsymbol{k}^{l,h^\top}}}{\sum_{j=1}^t e^{\boldsymbol{q}^{l,h} \boldsymbol{R}_{\Theta, j-i} \boldsymbol{k}^{l,h^\top}}} = \frac{e^{\|\boldsymbol{q}^{l,h}\| \|\boldsymbol{k}^{l,h}\| \cos(\beta_{t-i})}}{\sum_{j=1}^t e^{\|\boldsymbol{q}^{l,h}\| \|\boldsymbol{k}^{l,h}\| \cos(\beta_{t-j})}}. \tag{27}$$

Suppose the norm of multiplication for query and key $\left\| \boldsymbol{q}^{l,h} \right\| \left\| \boldsymbol{k}^{l,h} \right\| = \xi$. Considering $-1 \le \cos(\beta_{t-j}) \le 1$, then we have

$$\boldsymbol{A}_{t,i}^{l,h} = \frac{e^{\xi \cos(\beta_{t-i})}}{\sum_{j=1}^t e^{\xi \cos(\beta_{t-j})}} = \frac{1}{1 + \frac{\sum_{j \ne i} e^{\xi \cos(\beta_{t-j})}}{e^{\xi \cos(\beta_{t-i})}}} \le \frac{e^{2\xi}}{e^{2\xi} + (t-1)} \tag{28}$$

Then the attention scores for each token are upper-bounded and decrease to $0$ as $t$ grows. $\qquad\square$

For LMs with absolute PE/learnable PE, the initial hidden states $\boldsymbol{h}_t^0 = \boldsymbol{x} \boldsymbol{W}_E + \boldsymbol{p}_t$. Although the word embeddings are the same for repeated tokens, $\boldsymbol{p}_t$ for different token positions $t$ is different. Therefore, GPT2 models have no the above equality. From Table 1(*Left*), GPT2-XL still allocates significant attention to the first token even with repeated tokens, which motivates us to explore whether attention sink is related to these learned positional embedding vectors $\boldsymbol{p}_{1:T}$ after LM pre-training.

Therefore, we conduct two experiments on GPT2-XL. Firstly, we replace the first positional embedding vector $\boldsymbol{p}_1$ with other vectors $\boldsymbol{p}_{t \ne 1}$. In Table 7, we find that the amplitude of attention sink on the first token is significantly reduced. Then we also consider swapping the first positional embedding vector $\boldsymbol{p}_1$ with another position $\boldsymbol{p}_{t \ne 1}$. Consequently, the $t$-th token becomes the new sink token. Therefore, attention sink in GPT2-XL is strongly attached to the first positional embedding vector $\boldsymbol{p}_1$.

Table 7: In GPT2-XL, replacing or swapping the first positional embedding vector $\boldsymbol{p}_1$ with another position $\boldsymbol{p}_{t \neq 1}$ significantly impact the amplitude and position of attention sink.

| Replaced position $t$ | no | 5 | 10 | 15 | 20 | 25 | 25 |
|---|---|---|---|---|---|---|---|
| $\text{Sink}_1^\epsilon(\%)$ | 62.28 | 0.20 | 2.36 | 7.73 | 10.63 | 10.97 | 10.21 |
| $\text{Sink}_t^\epsilon(\%)$ | - | 0.00 | 0.00 | 0.00 | 0.00 | 0.00 | 0.00 |
| Swapped position $t$ | no | 5 | 10 | 15 | 20 | 25 | 25 |
| $\text{Sink}_1^\epsilon(\%)$ | 62.28 | 1.44 | 3.73 | 6.78 | 8.95 | 9.42 | 9.73 |
| $\text{Sink}_t^\epsilon(\%)$ | - | 57.63 | 54.48 | 52.81 | 51.70 | 51.13 | 50.22 |

## C.2 ATTENTION SINK UNDER DIFFERENT DATA DOMAINS

There are 17 available data domains in the Pile dataset (Gao et al., 2020), including Pile-CC, PubMed Central, ArXiv, Github, FreeLaw, Stack Exchange, USPTO Backgrounds, Pubmed Abstracts, Gutenberg (PG-19), Wikipedia (en), DM Mathematics, Ubuntu IRC, EuroParl, HackerNews, PhilPapers, NIH ExPorter, and Enron Emails. We sample 100 data from each domain and then evaluate the attention sink metric for GPT2-XL/Mistral-7B/LLaMA2-7B Base/LLaMA3-8B base. As shown in Figure 9, the evaluated attention sink metrics $\text{Sink}_1^\epsilon$ are similar across different domains when $\epsilon = 0.2$ and $\epsilon = 0.3$. Small fluctuations appear when $\epsilon = 0.4$.

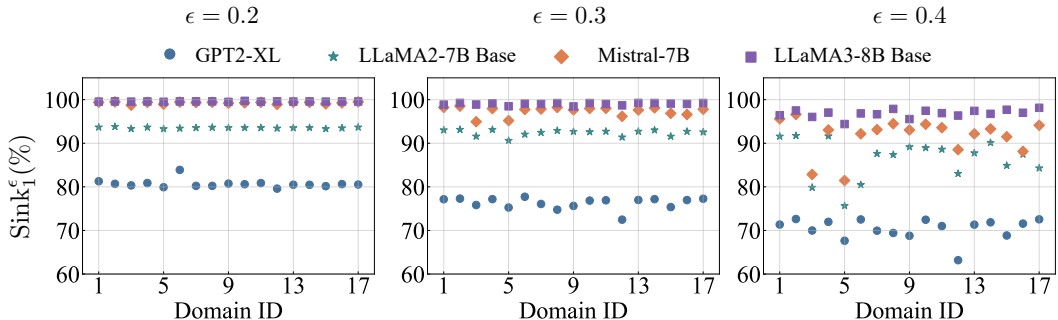

Figure 9: Input domains have negligible effects on attention sink metric $\text{Sink}_1^\epsilon$ for (*left*) $\epsilon = 0.2$ and (*middle*) $\epsilon = 0.3$. There are small fluctuations for (*right*) $\epsilon = 0.4$.

## C.3 ATTENTION SINK UNDER DIFFERENT PRE-TRAINED LMS

**Relation to LM performance.** We leverage the platform (Gao et al., 2024) to evaluate the performance of open-sourced LMs, including LLaMA2/LLaMA3/OPT/Pythia/GPT2 families, on downstream LM task, e.g., HellaSwag (Zellers et al., 2019). The results are visualized parallel with our attention sink metric in Figure 10, including both accuracy (Acc) and accuracy under normalization (Acc_Norm). We find that within the same LM family, with the increase of model scale, both attention sink amplitude and downstream LM performance are increasing. However, across different LM families, stronger attention sink does not always correlate with better performance. For instance, the OPT family has stronger attention sink than Pythia under the comparable model scale. However, the downstream LM performance is comparable.

$\ell_2$-**norm.** We first show that large $\ell_2$-norm of hidden states $\boldsymbol{h}_1^l$ and small $\ell_2$-norm of keys $\boldsymbol{k}_1^{l,h}$ and values $\boldsymbol{v}_1^{l,h}$ (especially for values) universally exist in open-sourced LMs, including LLaMA2-7B Base (Figure 11), GPT2-Large (Figure 12), Mistral-7B (Figure 13), and Pythia-1B (Figure 14). It is noted that for the final transformer block $l = L$, we take the hidden states before LN. We note that different LMs may have different starting blocks where massive activations appear.

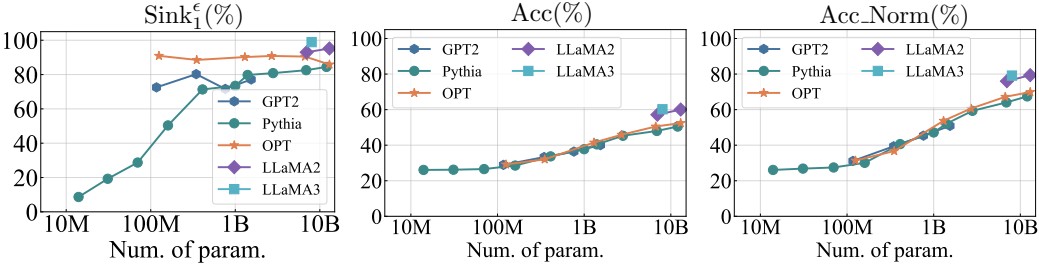

Figure 10: Attention sink and downstream performance for various pre-trained LMs.

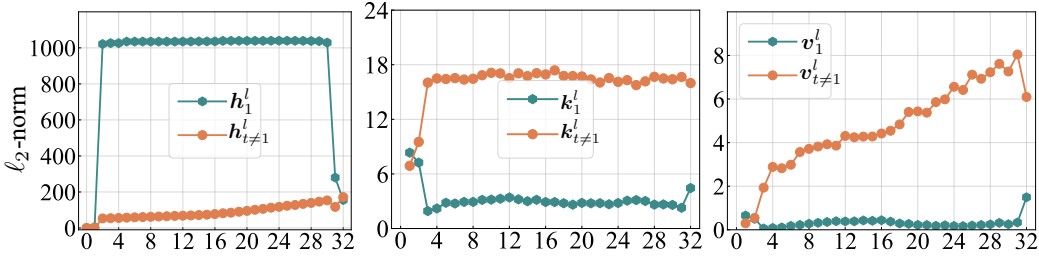

Figure 11: $\ell_2$-norm of hidden states/keys/values of the first token/other tokens in LLaMA2-7B Base.

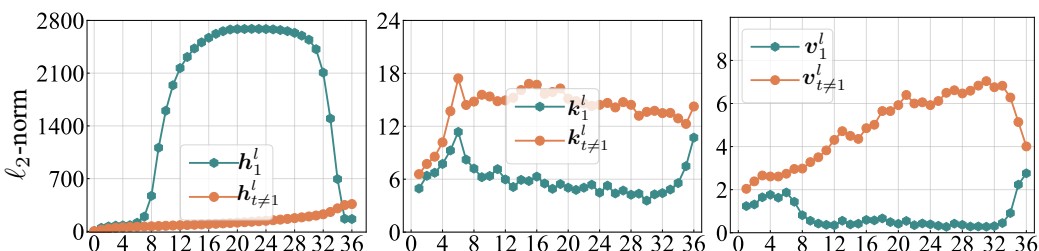

Figure 12: $\ell_2$-norm of hidden states/keys/values of the first token/other tokens in GPT2-Large.

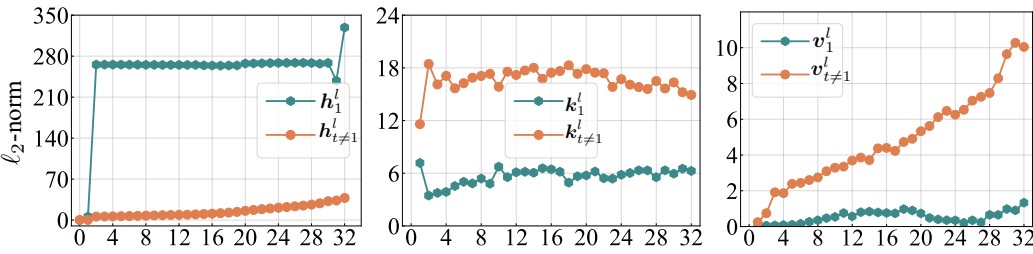

Figure 13: $\ell_2$-norm of hidden states/keys/values of the first token/other tokens in Mistral-7B.

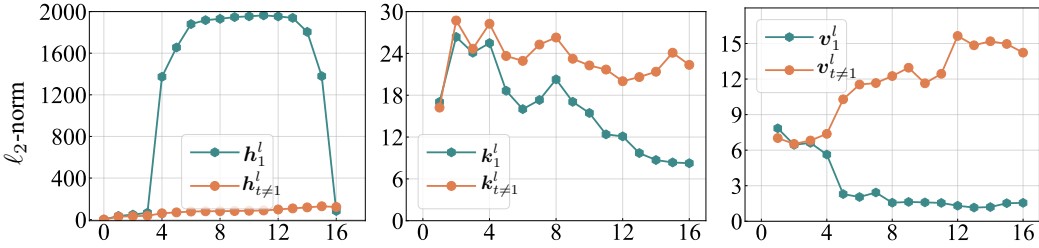

Figure 14: $\ell_2$-norm of hidden states/keys/values of the first token/other tokens in Pythia-1B.

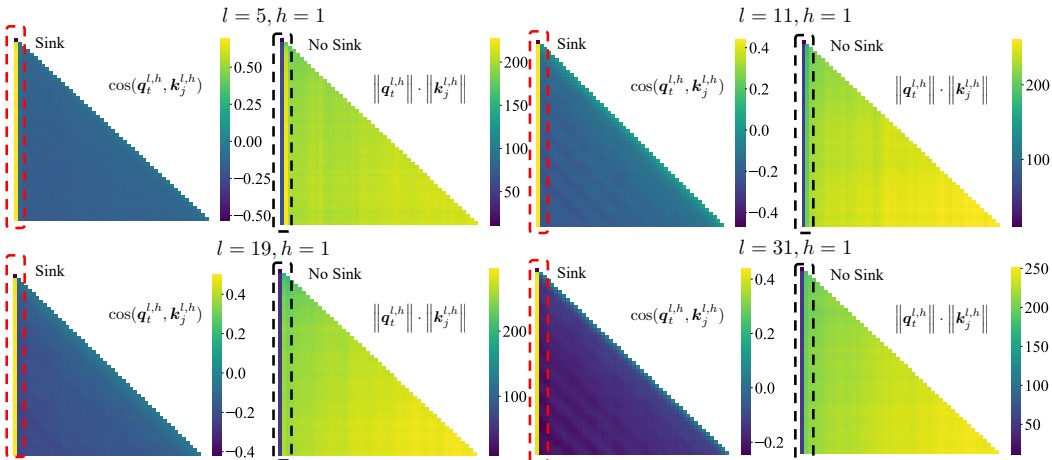

Figure 15: Cosine similarity and $\ell_2$-norm product between keys and queries in LLaMA3-8B Base.

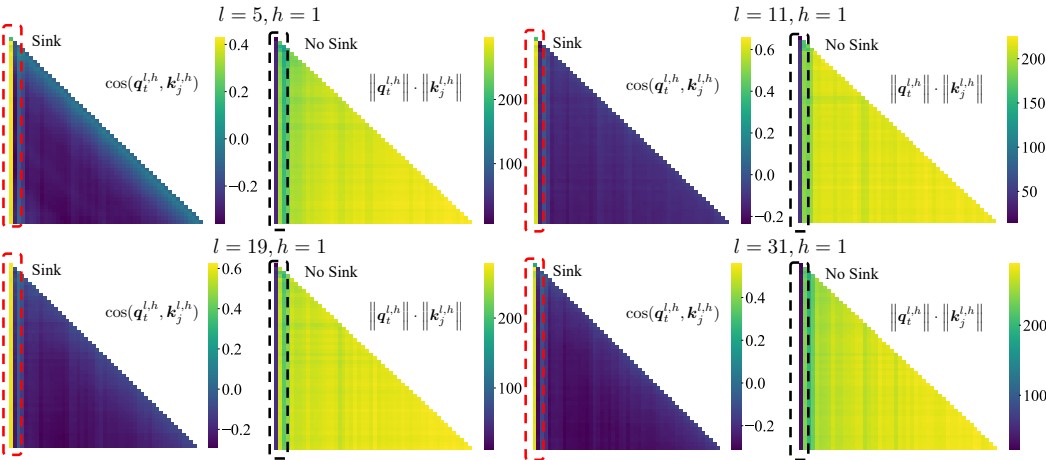

Figure 16: Cosine similarity and $\ell_2$-norm product between keys and queries in LLaMA2-7B Base.

**QK angles.** Then we further demonstrate that QK angles contribute to attention sink through more visualizations, including LLaMA3-8B Base (Figure 15), LLaMA2-7B Base (Figure 16), Mistral-7B (Figure 17), and GPT2-Large (Figure 18).

**Block-wise and head-wise property.** In the main paper, we mainly discuss the ratio of heads that have attention sink in the definition of attention sink metric $\text{Sink}_k^\epsilon$. Here we visualize the locations of these attention sink heads in open-sourced LMs, including LLaMA2 family (Figure 19), LLaMA3/LLaMA3.1 family (Figure 20), Mistral family (Figure 21), GPT2 family (Figure 22), Pythia family (Figure 23), and OPT family (Figure 24). We visualize the distributions of importance scores for the first token $\alpha_1^{l,h}$ across different transformer blocks $1 \leq l \leq L$ and different heads $1 \leq h \leq H$ before computing the attention sink metric. We find that (1) different pre-trained LMs have various attention sink distributions but they tend to have less obvious attention sink in earlier transformer blocks; (2) instruction tuning does not significantly modify such attention sink distributions when comparing base versions and chat/instruct versions.

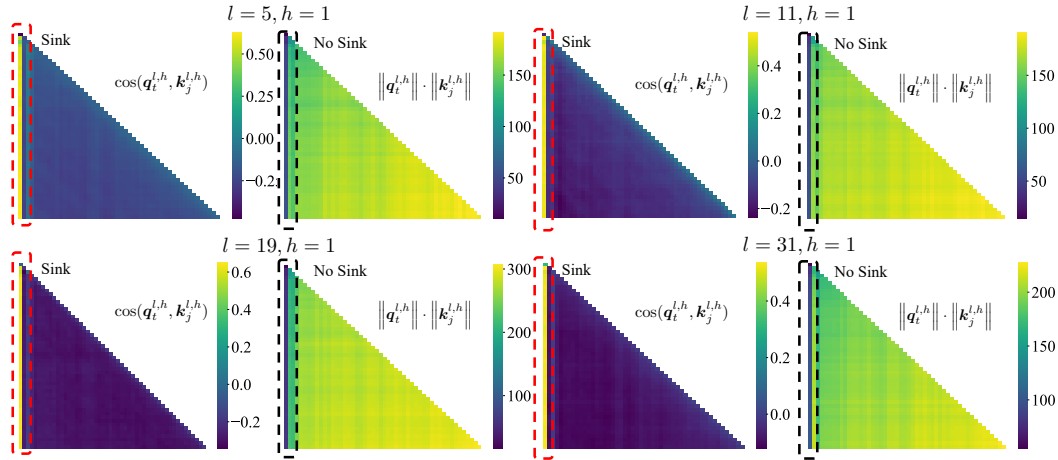

Figure 17: Cosine similarity and $\ell_2$-norm product between keys and queries in Mistral-7B.

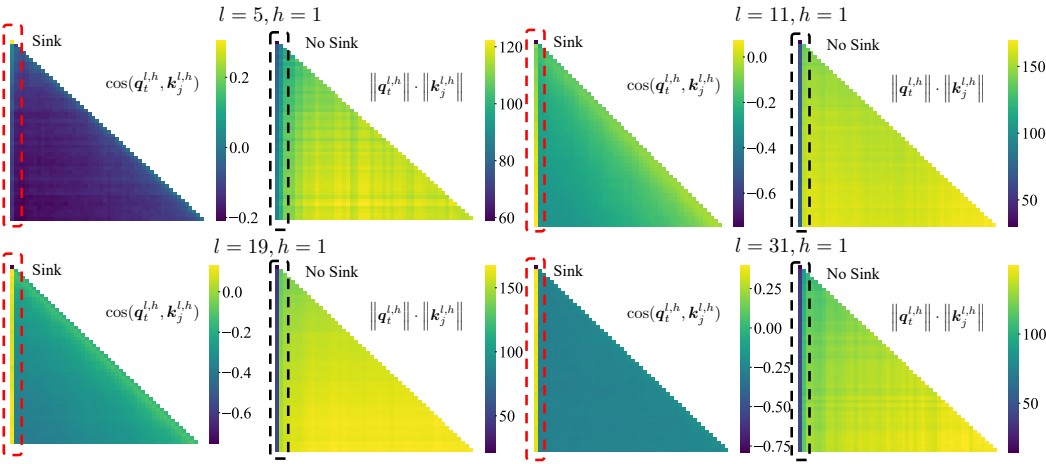

Figure 18: Cosine similarity and $\ell_2$-norm product between keys and queries in GPT2-Large.

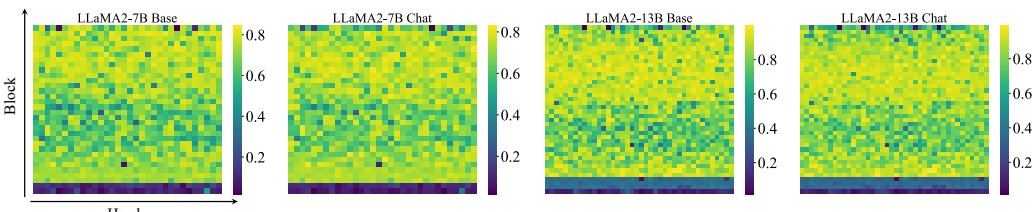

Figure 19: Distribution of importance scores for the first token across different blocks and heads in the LLaMA2 family.

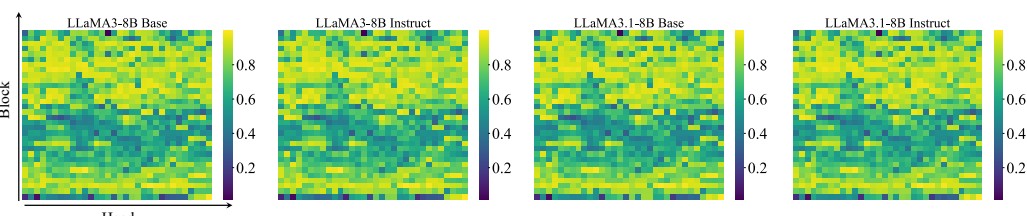

Figure 20: Distribution of importance scores for the first token across blocks and heads in the LLaMA3/LLaMA3.1 family.

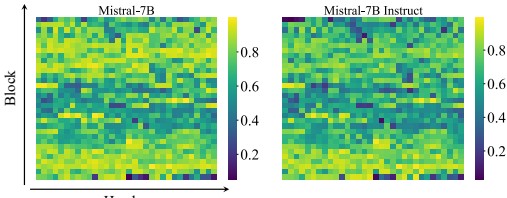

Figure 21: Distribution of importance scores for the first token across blocks and heads in the Mistral family.



Figure 22: Distribution of importance scores for the first token across blocks and heads in the GPT2 family.

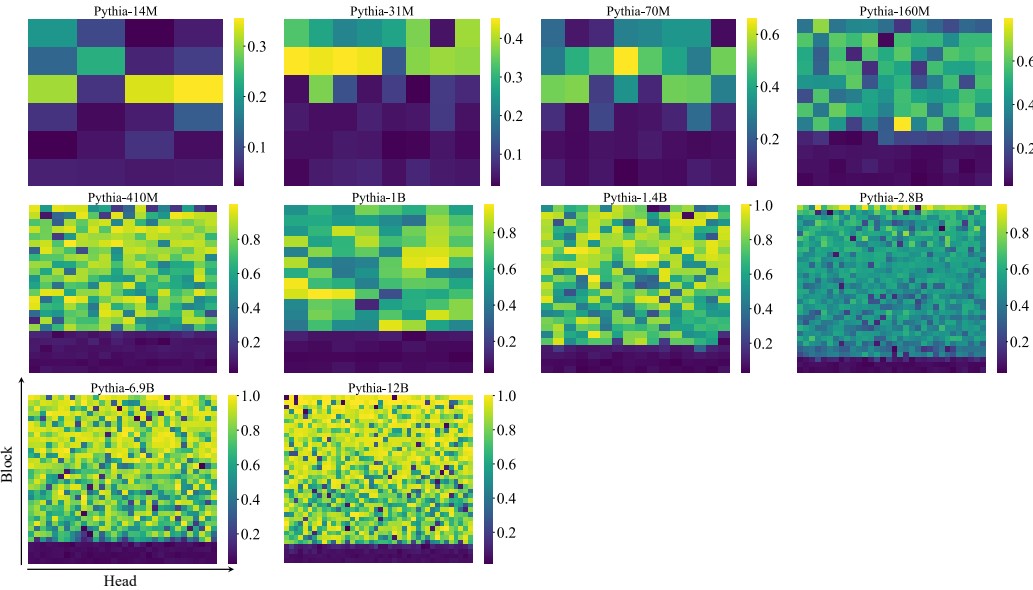

Figure 23: Distribution of importance scores for the first token across blocks and heads in the Pythia family.

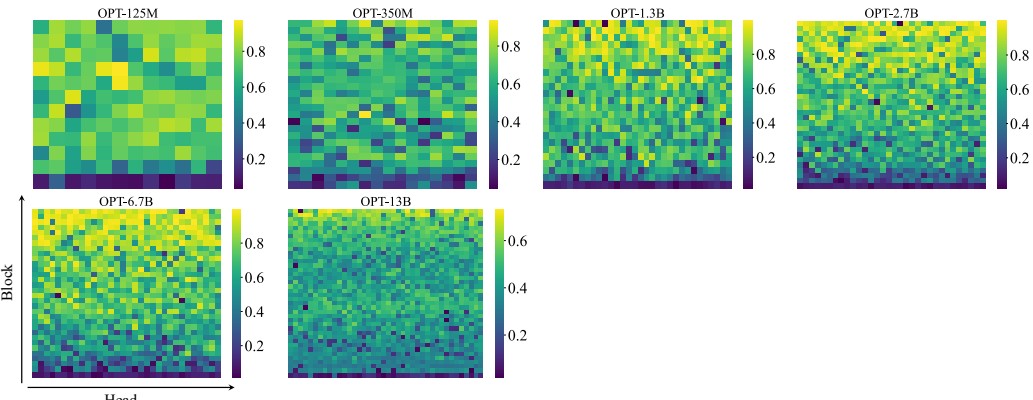

Figure 24: Distribution of importance scores for the first token across blocks and heads in the OPT family.

**Jamba.** Besides the auto-regressive Transformers, we also consider Jamba (Lieber et al., 2024; Team et al., 2024), a new foundation language model. Both Jamba-v0.1 (Lieber et al., 2024) and Jamba-1.5 Mini (Team et al., 2024) have 4 Jamba blocks, each of which includes 3 Mamba layers (Gu & Dao, 2023), 4 Mamba MoE layers (Shazeer et al., 2017; Fedus et al., 2022), and 1 Transformer layer. This adds to 32 layers (including 4 Transformer attention layers), 52B available parameters, and 12B active parameters in total. Firstly, we evaluate the attention sink metric and find that $\text{Sink}_1^{\epsilon} = 88.48\%$ for Jamba-v0.1 and $\text{Sink}_1^{\epsilon} = 87.88\%$ for Jamba-1.5 Mini, which indicates a strong attention sink on the first token. Then we visualize attention scores in several heads, as shown in Figure 25 and Figure 26. We also visualize the distribution of importance scores for the first token across blocks and heads in Jamba models in Figure 27. We observe that most heads have obvious attention sink, except for several heads in the 3rd Transformer layer.

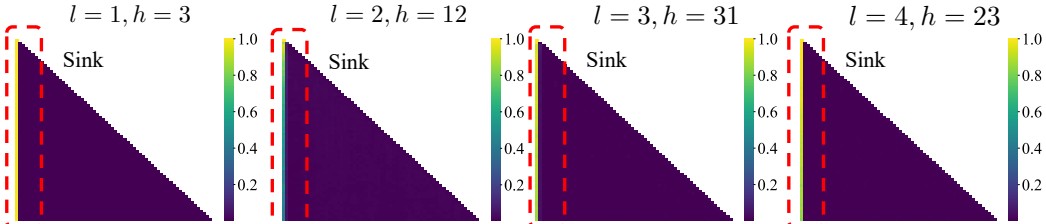

Figure 25: Attention sink in Jamba-v0.1.

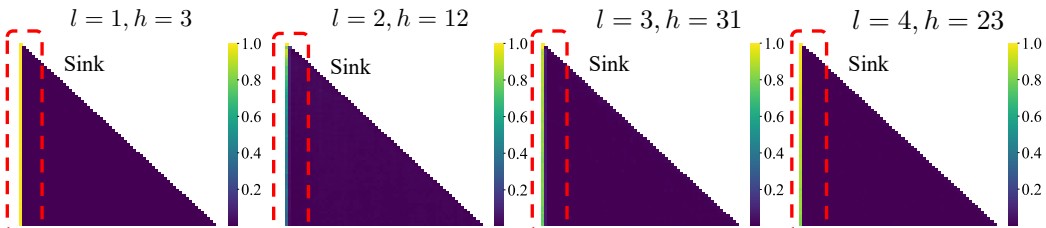

Figure 26: Attention sink in Jamba-1.5 Mini.

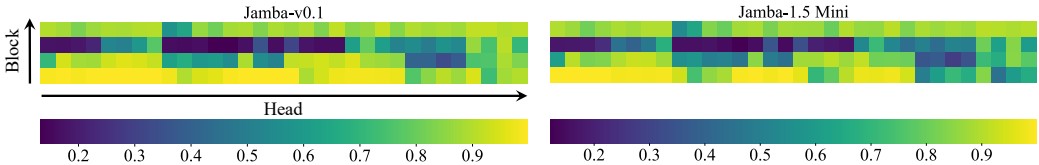

Figure 27: Distribution of importance scores for the first token across blocks and heads in the Jamba models.

## C.4    HUGGINGFACE LINKS FOR OPEN-SOURCED LMS

Table 8: Huggingface links for open-sourced LMs we used in this paper.

| Model | Huggingface link |
| --- | --- |
| LLaMA2-7B Base | meta-llama/Llama-2-7b-hf |
| LLaMA2-7B Chat | meta-llama/Llama-2-7b-chat-hf |
| LLaMA2-13B Base | meta-llama/Llama-2-13b-hf |
| LLaMA2-13B Chat | meta-llama/Llama-2-13b-chat-hf |
| LLaMA3-8B Base | meta-llama/Meta-Llama-3-8B |
| LLaMA3-8B Instruct | meta-llama/Meta-Llama-3-8B-Instruct |
| LLaMA3.1-8B Base | meta-llama/Meta-Llama-3.1-8B |
| LLaMA3.1-8B Instruct | meta-llama/Meta-Llama-3.1-8B-Instruct |
| GPT2 | openai-community/gpt2 |
| GPT2-Medium | openai-community/gpt2-medium |
| GPT2-Large | openai-community/gpt2-large |
| GPT2-XL | openai-community/gpt2-xl |
| Mistral-7B | mistralai/Mistral-7B-v0.1 |
| Mistral-7B Instruct | mistralai/Mistral-7B-Instruct-v0.1 |
| Pythia-14M | EleutherAI/pythia-14m |
| Pythia-31M | EleutherAI/pythia-31m |
| Pythia-70M | EleutherAI/pythia-70m |
| Pythia-160M | EleutherAI/pythia-160m |
| Pythia-410M | EleutherAI/pythia-410m |
| Pythia-1B | EleutherAI/pythia-1b |
| Pythia-1.4B | EleutherAI/pythia-1.4b |
| Pythia-2.8B | EleutherAI/pythia-2.8b |
| Pythia-6.9B | EleutherAI/pythia-6.9b |
| Pythia-12B | EleutherAI/pythia-12b |
| OPT-125M | facebook/opt-125m |
| OPT-350M | facebook/opt-350m |
| OPT-1.3B | facebook/opt-1.3b |
| OPT-2.7B | facebook/opt-2.7b |
| OPT-6.7B | facebook/opt-6.7b |
| OPT-13B | facebook/opt-13b |
| Jamba-v0.1 | ai21labs/Jamba-v0.1 |
| Jamba-1.5 Mini | ai21labs/AI21-Jamba-1.5-Mini |

# D MORE EXPERIMENTS IN LM PRE-TRAINING

## D.1 OPTIMIZATION

**Learning rate.** In Section 4, we find that attention sink appears less obvious in LMs trained with small learning rates. This conclusion holds even if we compensate for more training steps. In our basic setup, we adopt a learning rate of 4e-4 for 20k training steps. When we scale the learning rate to half, i.e., 2e-4, we scale the training steps to 2 times, i.e., 40k. As shown in Table 9, when we keep the multiply between learning rate and training steps constant (highlighted using cyan color), LMs trained with smaller learning rates tend to exhibit less obvious attention sink. Therefore, we conclude that small learning rates not only slow down the increase of attention sink but also mitigate attention sink even with longer training durations.

Table 9: Attention sink appears less obvious in LMs trained with small learning rates even compensating for more training steps.

| learning rate | training steps (k) | $\text{Sink}_1^\epsilon(\%)$ | valid loss |
|:---:|:---:|:---:|:---:|
| 8e-4 | 10 | 23.44 | 3.79 |
| 8e-4 | 20 | 32.23 | 3.70 |
| 4e-4 | 20 | 18.18 | 3.73 |
| 2e-4 | 20 | 11.21 | 3.78 |
| 2e-4 | 40 | 16.81 | 3.68 |
| 1e-4 | 20 | 2.90 | 3.92 |
| 1e-4 | 80 | 6.29 | 3.67 |

**Batch size.** During the pre-training, we consider different batch sizes with other hyper-parameters fixed. As shown in Table 10(*Left*), batch size does not affect the emergence of attention sink.

## D.2 DATA DISTRIBUTION

**Training data amount.** In Section 5, we find that with less training data, attention sink also disappears. Meanwhile, LMs are also prone for overfitting. To disentangle the effects of training data amount and overfitting on attention sink, we monitor the dynamics of train/valid loss and attention sink metric during the LM pre-training in a more granular level, as present in Figure 28. With only 50M and 100M training data, LMs overfit at very early stages, between 1k and 2k steps. Meanwhile, $\text{Sink}_1^\epsilon$ maintains a very small value (less than 1%). While for the setup of 5B training data, $\text{Sink}_1^\epsilon$ keeps increasing after a certain step. This indicates that the amount of training data, instead of overfitting, plays an important role in the emergence of attention sink.

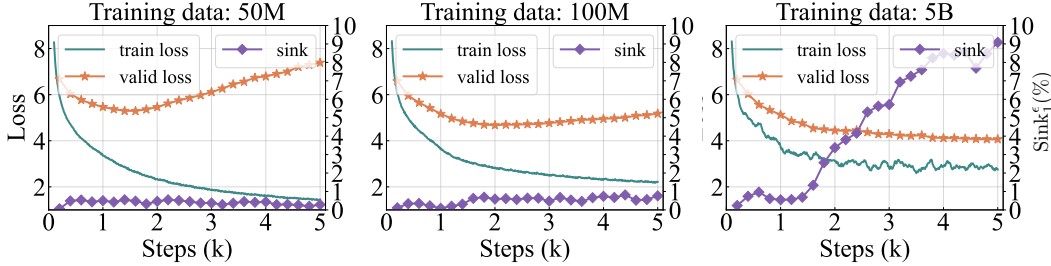

Figure 28: Dynamics of train/valid loss and $\text{Sink}_1^\epsilon$ during LM pre-training under different amounts of training data: (*Left*) 50M; (*Middle*) 100M; (*Right*) 5B.

**Fixing token in a specific position.** During the pre-training, we consider fixing the token $x_{\text{fix}}$ in the first/second/third position. Consequently, when evaluating the attention sink metric, in Table 10(*Right*), we find that attention sink appears in the appears in the fixed token instead of the first token.

Table 10: (*Left*) Batch size has no effect on the emergence of attention sink. (*Right*) Attention sink appears in the appears in the fixed token instead of the first token.

| Batch size | 0.25M | 0.5M | 1M | 2M |
|---|---|---|---|---|
| $\text{Sink}_1^\epsilon(\%)$ | 16.45 | 17.19 | 18.18 | 16.19 |
| valid loss | 3.92 | 3.78 | 3.73 | 3.68 |

| Fixed position | 1 | 2 | 3 |
|---|---|---|---|
| $\text{Sink}_1^\epsilon$ (%) | 74.11 | 0.00 | 0.00 |
| $\text{Sink}_2^\epsilon$ (%) | 0.00 | 69.03 | 0.00 |
| $\text{Sink}_3^\epsilon$ (%) | 0.00 | 0.00 | 69.64 |
| $\text{Sink}_4^\epsilon$ (%) | 0.01 | 0.01 | 0.00 |

## D.3 FFN DESIGN

**Activation functions.** Since FFN in the earlier transformer block blasts off the $\ell_2$-norm of the first token, we are wondering whether the choices of activation functions in FFN will affect the attention sink. Besides the SwiGLU activations used in our default setup, we also consider other activation functions, as present in Table 11. We observe that different FFN designs do not affect the emergence of attention sink.

Table 11: Modifying the activation functions in FFN does not affect the emergence of attention sink.

| Activation functions | $\boldsymbol{F}$ | $\text{Sink}_1^\epsilon(\%)$ | valid loss |
|---|---|---|---|
| ReLU | $\text{ReLU}(\boldsymbol{h}\boldsymbol{W}_1)\boldsymbol{W}_2$ | 16.90 | 3.82 |
| GeLU (Hendrycks & Gimpel, 2016) | $\text{GeLU}(\boldsymbol{h}\boldsymbol{W}_1)\boldsymbol{W}_2$ | 14.76 | 3.79 |
| Swish (Ramachandran et al., 2017) | $\text{Swish}(\boldsymbol{h}\boldsymbol{W}_1)\boldsymbol{W}_2$ | 17.89 | 3.80 |
| ReGLU (Shazeer, 2020) | $(\text{ReLU}(\boldsymbol{h}\boldsymbol{W}_1) \odot \boldsymbol{h}\boldsymbol{W}_2)\boldsymbol{W}_3$ | 13.88 | 3.75 |
| GeGLU (Shazeer, 2020) | $(\text{GeLU}(\boldsymbol{h}\boldsymbol{W}_1) \odot \boldsymbol{h}\boldsymbol{W}_2)\boldsymbol{W}_3$ | 17.86 | 3.73 |
| SwiGLU (Shazeer, 2020) | $(\text{Swish}(\boldsymbol{h}\boldsymbol{W}_1) \odot \boldsymbol{h}\boldsymbol{W}_2)\boldsymbol{W}_3$ | 18.18 | 3.73 |

## D.4 ATTENTION DESIGN

**Multi-head design in attention.** We consider two perspectives in multi-head design in attention. Firstly, we explore whether the number of heads has an impact on attention sink, especially for $H = 1$, which refers to single-head self-attention. Second, attention output from each head is concatenated: $\boldsymbol{O}^l = \text{Concat}_{h=1}^H \left(\boldsymbol{A}^{l,h}\boldsymbol{V}^{l,h}\right)\boldsymbol{W}_O^l$. We replace such concatenation operation with addition operation: $\boldsymbol{O}^l = \sum_{h=1}^H \left(\boldsymbol{A}^{l,h}\boldsymbol{V}^{l,h}\right)\boldsymbol{W}_O^l$. As shown in Table 12(*Left*), multi-head design does not affect the emergence of attention sink.

Table 12: (*Left*) Modifying multi-head design does not affect the emergence of attention sink. (*Right*) Sharing KV biases across heads in each block results in attention sink shifting back to the first token from K biases.

| Multi-head | $\text{Sink}_1^\epsilon$ (%) | valid loss |
|---|---|---|
| $H = 8$ | 18.18 | 3.73 |
| $H = 4$ | 10.68 | 3.74 |
| $H = 2$ | 12.95 | 3.76 |
| $H = 1$ | 19.50 | 3.78 |
| addition | 21.76 | 3.74 |

| Head-sharing | ✓ | ✓ | × | × |
|---|---|---|---|---|
| Biases type | KV | KV | K | K |
| $\text{Sink}_*^\epsilon(\%)$ | 72.76 | 56.61 | 73.34 | 68.31 |
| $\text{Sink}_1^\epsilon(\%)$ | 0.04 | 12.44 | 0.00 | 0.23 |
| valid loss | 3.72 | 3.72 | 3.72 | 3.72 |

**Head-sharing K/KV biases in attention.** In the main paper, we consider both KV biases and V biases in attention. These biases are not shared by different heads in each block. We further explore whether head-sharing patterns will affect their functionality. As present in Table 12(*Right*), LMs with KV biases are more likely affected by the head-sharing pattern: attention sink shifts from the K biases to the first token. While LMs with K biases are less affected.

**Learnable dimensions of K biases.** In the setup of K biases, we set all dimensions of $\boldsymbol{k}^{*l,h}$ as learnable weights. In Section 3.1, we have shown that K biases are distributed in a different manifold,

Table 13: Even with very few learnable dimensions for $\boldsymbol{k}^{*l,h}$, large attention appears in $\boldsymbol{k}^{*l,h}$.

| $d_a$ | 1 | 2 | 4 | 8 | 16 | 32 | 64 |
|---|---|---|---|---|---|---|---|
| $\text{Sink}_*^\epsilon(\%)$ | 32.18 | 30.88 | 30.94 | 31.39 | 23.30 | 51.23 | 69.19 |
| $\text{Sink}_1^\epsilon(\%)$ | 4.74 | 4.96 | 4.39 | 4.54 | 2.19 | 1.94 | 0.04 |
| valid loss | 3.73 | 3.72 | 3.72 | 3.73 | 3.73 | 3.73 | 3.72 |

with low rank. Therefore, we consider only $d_a$ dimensions of $\boldsymbol{k}^{*l,h}$ are adjustable/learnable while the other $d_h - d_a$ dimensions are zeros. As present in Table 13, with very few learnable dimensions, even for $d_a = 1$, $\boldsymbol{k}^{*l,h}$ are still allocated significant attention. With more learnable dimensions, the attention sink appears more obvious.

**Scaling the normalization in softmax attention.** Before our experiments about replacing softmax attention, we first explore the effects of normalization scales in softmax attention. In the main paper, the attention output for the $i$-th output is

$$\boldsymbol{v}_i^\dagger = \sum_{j=1}^i \frac{\text{sim}(\varphi(\boldsymbol{q}_i), \varphi(\boldsymbol{k}_j))}{\sum_{j'=1}^i \text{sim}(\varphi(\boldsymbol{q}_i), \varphi(\boldsymbol{k}_{j'}))} \boldsymbol{v}_j = \sum_{j=1}^i \frac{\text{sim}(\varphi(\boldsymbol{q}_i), \varphi(\boldsymbol{k}_j))}{\boldsymbol{Z}_i} \boldsymbol{v}_j. \tag{29}$$

We consider a scale factor $\alpha$, the normalization term is $\boldsymbol{Z}_i = \frac{1}{\alpha} \sum_{j'=1}^i \text{sim}(\varphi(\boldsymbol{q}_i), \varphi(\boldsymbol{k}_j))$, and then the attention score are sum up to $\alpha$. For default setup, $\alpha = 1$. As shown in Table 14(*Left*), with a smaller normalization scale, attention sink tends to appear in fewer heads. From another perspective,

$$\boldsymbol{v}_i^\dagger = \sum_{j=1}^i \frac{\alpha \text{sim}(\varphi(\boldsymbol{q}_i), \varphi(\boldsymbol{k}_j))}{\sum_{j'=1}^i \text{sim}(\varphi(\boldsymbol{q}_i), \varphi(\boldsymbol{k}_{j'}))} \boldsymbol{v}_j = \sum_{j=1}^i \frac{\text{sim}(\varphi(\boldsymbol{q}_i), \varphi(\boldsymbol{k}_j))}{\sum_{j'=1}^i \text{sim}(\varphi(\boldsymbol{q}_i), \varphi(\boldsymbol{k}_{j'}))} \boldsymbol{h}_j(\alpha \boldsymbol{W}_V), \tag{30}$$

$$\boldsymbol{o}_i' = \text{Concat}_{h=1}^H (\boldsymbol{v}_i'^h) \boldsymbol{W}_O. \tag{31}$$

Therefore, this normalization scale can be regarded as the scale for $\boldsymbol{W}_V$ or $\boldsymbol{W}_O$. We show that this normalization scaling could be implemented by scaling learning rates and initialization. We use the $s$ to represent the optimization step, and $s = 0$ refers to the initialization. When scaling the normalization, we have the following SGD update rule (take $\boldsymbol{W}_O$ for example):

$$\boldsymbol{W}_O^{s+1} = \boldsymbol{W}_O^s - \eta \nabla_{\boldsymbol{W}_O^s} \mathcal{L}(\alpha \boldsymbol{W}_O^s) \tag{32}$$

$$= \boldsymbol{W}_O^s - \alpha \eta \nabla_{\boldsymbol{W}} \mathcal{L}(\boldsymbol{W})|_{\boldsymbol{W}=\alpha \boldsymbol{W}_O^s}, \tag{33}$$

where $\eta$ is the original learning rate. Suppose we only modify the learning rate and initialization, to ensure each optimization step $\hat{\boldsymbol{W}}_O^s = \alpha \boldsymbol{W}_O^s$, we need first to ensure $\hat{\boldsymbol{W}}_O^0 = \alpha \boldsymbol{W}_O^0$. Suppose that we have $\hat{\boldsymbol{W}}_O^s = \boldsymbol{W}_O^s$, then the update rule is:

$$\hat{\boldsymbol{W}}_O^{s+1} = \hat{\boldsymbol{W}}_O^s - \eta' \nabla_{\hat{\boldsymbol{W}}_O^s} \mathcal{L}(\hat{\boldsymbol{W}}_O^s) \tag{34}$$

$$= \alpha \boldsymbol{W}_O^s - \eta' \nabla_{\boldsymbol{W}} \mathcal{L}(\boldsymbol{W})|_{\boldsymbol{W}=\alpha \boldsymbol{W}_O^s}, \tag{35}$$

To ensure $\hat{\boldsymbol{W}}_O^{s+1} = \alpha \boldsymbol{W}_O^{s+1}$, we need the new learning rate $\eta'$ meets $\eta' = \alpha^2 \eta$. For advanced optimization algorithms, e.g., Adam (Kingma & Ba, 2014) and AdamW (Loshchilov & Hutter, 2017). We have the following update rule (take AdamW for example, $\gamma$ refers to the weight decay ratio):

$$\boldsymbol{g}_{s+1} = \nabla_{\boldsymbol{W}_O^s} \mathcal{L}(\alpha \boldsymbol{W}_O^s) = \alpha \nabla_{\boldsymbol{W}} \mathcal{L}(\boldsymbol{W})|_{\boldsymbol{W}=\alpha \boldsymbol{W}_O^s} \tag{36}$$

$$\boldsymbol{m}_{s+1} = \beta_1 \boldsymbol{m}_s + (1 - \beta_1) \boldsymbol{g}_{s+1} \tag{37}$$

$$\boldsymbol{v}_{s+1} = \beta_2 \boldsymbol{v}_s + (1 - \beta_2) \boldsymbol{g}_{s+1}^2 \tag{38}$$

$$\boldsymbol{W}_O^{s+1} = (1 - \eta\gamma) \boldsymbol{W}_O^s - \eta \frac{\boldsymbol{m}_{s+1}/(1 - \beta_1^t)}{\sqrt{\boldsymbol{v}_{s+1}/(1 - \beta_2^t)} + \epsilon} \tag{39}$$

We denote $\hat{\boldsymbol{g}}_{s+1}$, $\hat{\boldsymbol{m}}_{s+1}$, $\hat{\boldsymbol{v}}_{s+1}$ represents the intermediate counterparts for update of scenario where we only modify learning rate and initialization. First, we also need to ensure the initialization $\hat{\boldsymbol{W}}_O^0 = \alpha \boldsymbol{W}_O^0$. Then we assume that we have already matched $\hat{\boldsymbol{W}}_O^s = \alpha \boldsymbol{W}_O^s$. The gradient for each step is

$$\hat{\boldsymbol{g}}_{s+1} = \nabla_{\hat{\boldsymbol{W}}_O^s} \mathcal{L}(\hat{\boldsymbol{W}}_O^s) = \nabla_{\boldsymbol{W}} \mathcal{L}(\boldsymbol{W})|_{\boldsymbol{W}=\alpha \boldsymbol{W}_O^s} = \boldsymbol{g}_{s+1}/\alpha \tag{40}$$

Then the first and second moment will be $\hat{m}_{s+1} = m_{s+1}/\alpha$ and $\hat{v}_{s+1} = v_{s+1}/\alpha^2$. The updated weights will be

$$\hat{W}_O^{s+1} = (1 - \eta'\gamma')\hat{W}_O^s - \eta'\frac{\hat{m}_{s+1}/(1 - \beta_1^t)}{\sqrt{\hat{v}_{s+1}/(1 - \beta_2^t)} + \epsilon'} \tag{41}$$

$$= (1 - \eta'\gamma')\alpha W_O^s - \eta'\frac{m_{s+1}/\alpha(1 - \beta_1^t)}{\sqrt{v_{s+1}/\alpha^2(1 - \beta_2^t)} + \epsilon'} \tag{42}$$

Therefore, to ensure $\hat{W}_O^{s+1} = \alpha W_O^{s+1}$, one solution is $\eta' = \alpha\eta$ and $\epsilon' = \epsilon/\alpha$ and $\gamma' = \gamma/\alpha$.

Table 14: (*Left*) Scaling the normalization in Softmax attention to less than one can mitigate attention sink but not prevent its emergence. (*Right*) LMs with sigmoid attention (without sigmoid attention) trained by different learning rates and weight decay ratios have no attention sink.

| Scale $\alpha$ | Sink$_1^\epsilon$(%) | valid loss | Learning rate | Weight decay | Sink$_1^\epsilon$(%) | valid loss |
|---|---|---|---|---|---|---|
| 2.0 | 29.10 | 3.72 | 4e-4 | 0.0 | 0.64 | 3.77 |
| 1.0 | 18.18 | 3.73 | 4e-4 | 0.1 | 0.44 | 3.70 |
| 0.2 | 9.41 | 3.72 | 4e-4 | 0.5 | 0.18 | 3.76 |
| 0.1 | 3.59 | 3.76 | 4e-4 | 1.0 | 0.30 | 4.06 |
| 0.05 | 4.53 | 3.78 | 1e-3 | 0.1 | 0.81 | 3.68 |
| | | | 1e-4 | 0.1 | 0.36 | 4.08 |

**Normalizer.** Besides the normalizer $Z_i = \sum_{j'=1}^i \text{sim}(\varphi(q_i), \varphi(k_{j'}))$ considered in the main paper, we consider alternative normalizer: $Z_i = \left(\sum_{j'=1}^i \text{sim}(\varphi(q_i), \varphi(k_{j'}))^p\right)^{\frac{1}{p}}$, which gives us following attention operation:

$$v_i^\dagger = \frac{\sum_{j=1}^i \text{sim}(\varphi(q_i), \varphi(k_j))v_j}{Z_i} = \frac{\sum_{j=1}^i \text{sim}(\varphi(q_i), \varphi(k_j))v_j}{\left(\sum_{j'=1}^i \text{sim}(\varphi(q_i), \varphi(k_{j'}))^p\right)^{\frac{1}{p}}}. \tag{43}$$

For softmax attention, we have $\text{sim}(\varphi(q_i), \varphi(k_j)) = \exp(\frac{q_i k_j^\top}{\sqrt{d_h}})$, then we can derive that

$$v_i^\dagger = \frac{\sum_{j=1}^i \exp(\frac{q_i k_j^\top}{\sqrt{d_h}}))v_j}{\left(\sum_{j'=1}^i \exp(\frac{q_i k_{j'}^\top}{\sqrt{d_h}})^p\right)^{\frac{1}{p}}} = \sum_{j=1}^i \left(\frac{\exp(\frac{q_i k_j^\top}{\sqrt{d_h}/p})}{\sum_{j'=1}^i \exp(\frac{q_i k_{j'}^\top}{\sqrt{d_h}/p})}\right)^{\frac{1}{p}} v_j. \tag{44}$$

This is equivalent to adding a temperature $1/p$ into the softmax attention logits, and then taking the $p$-root of attention scores after softmax. We call this $p$-normalized softmax attention. $p = 1$ in regular softmax attention. Similarly, we construct the LMs with $p$-normalized sigmoid attention.

We find that when $p = 2$ or $p = 3$ or $p = 4$, pre-training of $p$-normalized softmax attention diverges and the loss goes infinity. When $p = 1/2$ or $p = 1/3$ or $p = 1/4$, LM pre-training converges. As the attention scores are not added up to one, we investigate the massive activations instead, as visualized in Figure 29(*Left*). With smaller $p$, massive activations are mitigated to some extent, but not as effective as sigmoid attention without normalization. Intuitively, smaller $p$ induces a larger temperature in softmax operation, which leads to flattened attention logits.

Afterward, we conduct experiments on $p$-normalized sigmoid attention. There is no training problem with $p$ larger than 1. As visualized in Figure 29(*Right*), LMs with $p$-normalized sigmoid attention still demonstrate strong massive activations. To conclude, different normalizers may affect the amplitude of attention sink, but not stop its emergence.

**Attention operations.** Firstly, we present all attempted attention operations in Table 15. It is noted that several setups lead to training failure. For LMs with sigmoid attention without normalization, we vary the learning rates or weight decay ratios $\gamma$. Consequently, as shown in Table 14(*Right*), the trained LMs still have no attention sink, which further confirms our conclusion in the main paper.

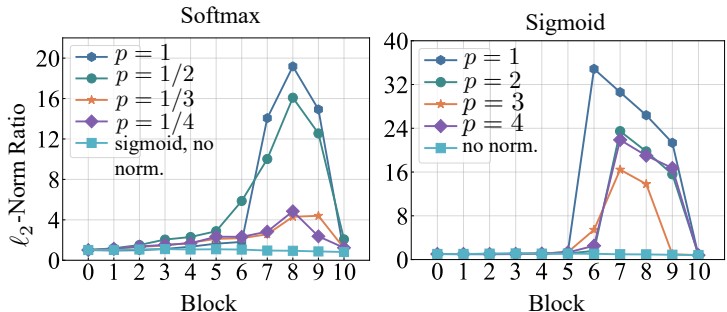

Figure 29: $\ell_2$-norm ratio of $\boldsymbol{h}_1^l$ and mean of $\boldsymbol{h}_{t\neq 0}^l$. (*Left*) $p$-normalized softmax attention and also sigmoid attention without normalization (as reference). (*Right*) $p$-normalized sigmoid attention and also sigmoid attention without normalization (as reference).

Table 15: Normalization and selections of kernels in attention significantly affect the emergence of the attention sink. We use "*" to mark that the metric $\text{Sink}_1^\epsilon$ is computed by proxy attention scores. We use "-" to represent the training failure under the setup.

| $\text{sim}(\varphi(\boldsymbol{q}_i), \varphi(\boldsymbol{k}_j))$ | $\boldsymbol{Z}_i$ | $\text{Sink}_1^\epsilon(\%)$ | valid loss |
|---|---|---|---|
| $\exp(\frac{\boldsymbol{q}_i \boldsymbol{k}_j^\top}{\sqrt{d_h}})$ | $\sum_{j'=1}^i \exp(\frac{\boldsymbol{q}_i \boldsymbol{k}_{j'}^\top}{\sqrt{d_h}})$ | 18.18 | 3.73 |
| $\text{sigmoid}(\frac{\boldsymbol{q}_i \boldsymbol{k}_j^\top}{\sqrt{d_h}})$ | $1$ | 0.44* | 3.70 |
| $\text{sigmoid}(\frac{\boldsymbol{q}_i \boldsymbol{k}_j^\top}{\sqrt{d_h}})$ | $\sum_{j'=1}^i \text{sigmoid}(\frac{\boldsymbol{q}_i \boldsymbol{k}_{j'}^\top}{\sqrt{d_h}})$ | 30.24 | 3.74 |
| $\text{elu}(\frac{\boldsymbol{q}_i \boldsymbol{k}_j^\top}{\sqrt{d_h}}) + 1$ | $1$ | 0.80* | 3.69 |
| $\text{elu}(\frac{\boldsymbol{q}_i \boldsymbol{k}_j^\top}{\sqrt{d_h}}) + 1$ | $\sum_{j'=1}^i \text{elu}(\frac{\boldsymbol{q}_i \boldsymbol{k}_{j'}^\top}{\sqrt{d_h}}) + 1$ | - | - |
| $\frac{(\text{elu}(\boldsymbol{q}_i)+1)(\text{elu}(\boldsymbol{k}_j)+1)^\top}{\sqrt{d_h}}$ | $\sum_{j'=1}^i \frac{(\text{elu}(\boldsymbol{q}_i)+1)(\text{elu}(\boldsymbol{k}_{j'})+1)^\top}{\sqrt{d_h}}$ | 53.65* | 4.19 |
| $\frac{(\text{elu}(\boldsymbol{q}_i)+1)(\text{elu}(\boldsymbol{k}_j)+1)^\top}{\sqrt{d_h}}$ | $1$ | - | - |
| $\frac{\boldsymbol{q}_i \boldsymbol{k}_j^\top}{\sqrt{d_h}}$ | $\max\left(\left|\sum_{j'=1}^i \frac{\boldsymbol{q}_i \boldsymbol{k}_{j'}^\top}{\sqrt{d_h}}\right|, 1\right)$ | - | - |
| $\frac{\boldsymbol{q}_i \boldsymbol{k}_j^\top}{\sqrt{d_h}}$ | $1$ | 0.00* | 3.99 |
| $\frac{\text{mlp}(\boldsymbol{q}_i)\text{mlp}(\boldsymbol{k}_j)^\top}{\sqrt{d_h}}$ | $\max\left(\left|\sum_{j'=1}^i \frac{\text{mlp}(\boldsymbol{q}_i)\text{mlp}(\boldsymbol{k}_{j'})^\top}{\sqrt{d_h}}\right|, 1\right)$ | 0.19* | 3.85 |
| $\frac{\text{mlp}(\boldsymbol{q}_i)\text{mlp}(\boldsymbol{k}_j)^\top}{\sqrt{d_h}}$ | $1$ | 0.74* | 3.91 |

# E MORE EXPERIMENTS IN LM AFTER PRE-TRAINING

**Training stability in supervised fine-tuning.** To investigate the long-term impacts of attention sink on model behaviors after pre-training, we conduct supervised fine-tuning (SFT) on our pre-trained 1B LMs with softmax attention and sigmoid attention without normalization. Specifically, we utilize the platform[1] to conduct our experiments. The experimental configurations include: the UltraChat dataset (about 200k training samples)[2] (Ding et al., 2023), full-model fine-tuning, a learning rate of 2e-5 with cosine scheduling, batch size of 64, each of which contains 2048 tokens, one training epoch. As shown in Figure 30, we monitor the training loss and gradient norm of our two LMs during SFT. These two models behave similarly in terms of the above two metrics. Additionally, despite no attention sink, LMs with sigmoid attention without normalization have no issues of training stability during SFT. Though not from the attention sink perspective, a concurrent work by Ramapuram et al. (2024) discussed the theory and practices for Transformer models with sigmoid attention in detail. We refer the readers to Ramapuram et al. (2024) for more analyses.

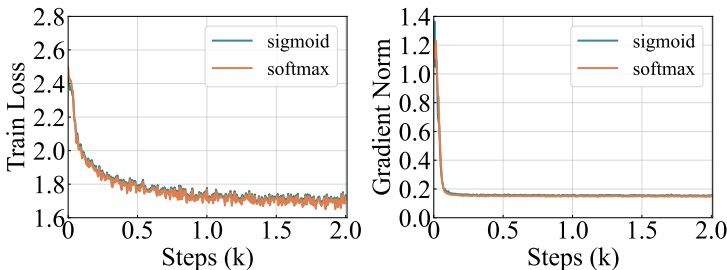

Figure 30: The training loss and gradient norm in our 1B LMs with softmax attention and sigmoid attention without normalization in supervised fine-tuning.

---

[1] https://github.com/huggingface/alignment-handbook
[2] https://huggingface.co/datasets/HuggingFaceH4/ultrachat_200k

