# OpenReview forum: "When Attention Sink Emerges in Language Models: An Empirical View"
_ICLR.cc/2025/Conference — ICLR 2025 Spotlight_

### Official Review · Reviewer_huvS · 2024-10-28

**Soundness:** 4
**Presentation:** 3
**Contribution:** 4
**Rating:** 8
**Confidence:** 5

**Summary:**

This paper studies the phenomenon of "attention sink" in Language Models, where significant attention is disproportionately allocated to the first token in input sequences, without taking into account its semantic importance. The attention sink has practical applications in areas such as streaming generation, KV cache optimization, inference acceleration, and model quantization.

The authors performed a study across various LM architectures, input types, and training configurations to study the emergence and characteristics of attention sinks.
They found that attention sink happens across different LMs, including smaller models and those trained on random token sequences and that it emerges during optimization with sufficient training data, and its prominence is influenced by learning rates and weight decay.
They also found that attention sink functions similarly to key biases, storing additional attention scores that are largely non-informative and do not contribute to value computations. This behavior is partially attributed to the softmax normalization, which induces inner dependencies among attention scores.

The paper also shows that altering the attention mechanism, such as replacing softmax with sigmoid attention without normalization, can prevent the emergence of attention sinks in LMs up to 1 billion parameters.

**Strengths:**

- Reproducibily.
- The paper is well written and the experiments set up is soild.
- The paper not only observes the presence of attention sinks but also delves into the mechanistic reasons behind their emergence (also providing insights on the training dynamics that foster attention sinks).
- The paper show that attention sinks persist across various input types, like random token sequences and repeated tokens.

I think that this paper is a valuable step further in understanding the phenomenon of attention sink.

**Weaknesses:**

- The paper doesn't really take into account the role of attention sink impact on downstream tasks.
- While the paper studies the emergence and immediate characteristics of attention sinks, it does not assess the long-term impacts of attention sinks on model behavior (like stability during fine-tuning, adaptability to new tasks, or resistance to adversarial attacks).

Not really a weakness, but i found the paper a bit difficoult to read because of the unusual aesthetic choices: i think that this style would be great for a blog post where you have more space available to write, but i personally find easier to follow simpler-looking papers.

**Questions:**

In your paper you study how to mitigate the attention sink, are you sure that we want to mitigate it?

---

> ### Author Response · Authors · 2024-11-19
> **Rebuttal by Authors**
>
> Thank you for your supportive review and suggestions. Below we respond to the comments in **Weaknesses (W) and Questions (Q)**.
>
> ---
>
> ***W1: The paper doesn't really take into account the role of attention sink impact on downstream tasks.***
>
> Thank you for your suggestions. We evaluate the performance of various pre-trained LMs using HellaSwag, a benchmark dataset for LMs. Afterward, we visualize the attention sink metric, accuracy, and normalized accuracy in $\\textrm{\\color{blue}Figure 10}$ (page 19) for comparative analysis. Within the same LM family, we observe that an increase in model scale correlates with improved performance on downstream tasks. Nonetheless, within the OPT family and the GPT2 family, our attention sink metric indicates a similar level across different model scales. Additionally, the OPT family has stronger attention sink than Pythia at comparable model scales, while its performance on downstream task performance is similar.
>
> ---
>
> ***W2: The paper does not assess the long-term impacts of attention sinks on model behavior (like stability during fine-tuning, adaptability to new tasks, or resistance to adversarial attacks).***
>
> In the paper revision, we conduct additional experiments concerning the stability during supervised fine-tuning (SFT) of our pre-trained 1B models, which include one employing standard softmax attention (exhibiting attention sink) and another utilizing sigmoid attention without normalization (lacking attention sink). We fully fine-tune these two models on the UltraChat dataset, adhering to a well-adopted training recipe. $\\textrm{\\color{blue}Figure 30}$ (page 30) illustrates the dynamics of training loss and gradient norm for our two LMs during SFT. These two models exhibit comparable performance for the aforementioned metrics. Furthermore, despite the absence of an attention sink, language models employing sigmoid attention without normalization exhibit no training stability difficulties during supervised fine-tuning.
>
> Due to the time limit during rebuttal, we will conduct more comprehensive experiments to explore the long-term impacts of attention sinks on model behavior in the final revision.
>
> ---
>
> ***W3: Not really a weakness, but I found the paper a bit difficult to read because of the unusual aesthetic choices.***
>
> Thank you for raising this. In the final revision, we will take more consideration about the readability of our paper.
>
> ---
>
> ***Q1: In your paper you study how to mitigate the attention sink, are you sure that we want to mitigate it?***
>
> Thank you for such an insightful question. We emphasize whether we want to mitigate attention sink remains an open and non-trivial issue. Attention sink indeed has numerous beneficial applications in practice, such as streaming generation, KV cache optimization, efficient inference, and model quantization. When there is attention sink, we appear to have guidelines on how to save memory and computational costs during inference. However, these approaches are still somewhat ad-hoc. If we could mitigate attention sink during the pre-training, we may obtain LMs that are naturally less redundant. We leave how to train such LMs (model architecture, optimization algorithm, etc.) to future work.
>
> Recently, the differential transformers [2] indicate that the design of multi-head differential attention mitigates attention sink / massive activations to some extent. In the meantime, the differential transformers outperform the baseline transformers regarding scaling properties, long-context modeling, hallucination mitigation, and in-context learning. Consequently, it is challenging to determine the necessity of attention sink in the next generation of foundation models. We believe this question will be fruitful for further exploration.
>
> ---
>
> ***References:*** \
> [1] Ding et al. Enhancing chat language models by scaling high-quality instructional conversations. EMNLP 2023\
> [2] Ye et al. Differentiable Transformer. arxiv 2024

---

> > ### Comment · Reviewer_huvS · 2024-11-20
> >
> > Thank you for your response.
> > I read also the other reviews and the authors' responses and the rebuttal is convicing and I especially enjoyed reading the authors' opinion about the need of mitigating the attention sink.
> > So, i'd like to retain my overall score of 8.

---

> > > ### Author Response · Authors · 2024-11-20
> > > **Thank you for your support**
> > >
> > > Thank you for your timely feedback and kind words. We really appreciate it! In the final revision, we will further polish our paper to incorporate the insights from the rebuttal discussions. Thank you again!

---

### Official Review · Reviewer_xfDZ · 2024-10-31

**Soundness:** 4
**Presentation:** 4
**Contribution:** 3
**Rating:** 8
**Confidence:** 4

**Summary:**

In this paper, the authors conduct a comprehensive study of the attention sink problem and present robust empirical results. They discuss and examine the attention sink problem from various perspectives, including optimization, data distribution, loss function, and model architecture. Although the paper does not provide in-depth theoretical analysis, it may inspire further research into the understanding of the attention mechanism, which could, in turn, contribute to the development of stronger generative models. Therefore, I believe this paper would serve as a valuable empirical reference on the attention sink problem for the community and worths of acceptance.

**Strengths:**

1. The perspectives of studying attention sink problem is diverse and well-motivated. These perspectives are also inspiring to future studies on attention mechanisms.
2. The experiments is very comprehensive and diverse.

**Weaknesses:**

The primary weakness of this paper is the lack of in-depth analysis. The empirical results come across more as observations rather than as thorough investigations. Including more theoretical analysis or deeper experimental work would strengthen the paper. For instance, in the KV bias section, it would be beneficial to explore how attention variants with and without attention sink are related in formulation.

**Questions:**

Q1: In Fig. 4 (left), could you provide a performance curve, such as the validation performance of each model against model size? It appears that as the model becomes stronger, attention sink becomes more prominent. Additionally, I am curious about how attention sink correlates with validation loss.

Q2: Fig. 4 (right) shows that a lower learning rate results in a slower increase in attention sink. Is it possible that attention sink occurs simply because the model has not been well-tuned?

Q3: In Table 1, GPT-XL behaves very differently from Llama and Mistral. Do you have any intuition as to why this might be?

Q4: Could you provide some mathematical formulation on how the attention variants (2nd to 4th rows in Table 4) are correlated? Is there an inherent connection among these attention variants that excludes attention sink?

Q5: In Table 6, it appears that the normalizer may impact attention sink. Do you have additional evidence with other types of normalizers?

---

> ### Author Response · Authors · 2024-11-19
> **Rebuttal by Authors [1/3]**
>
> Thank you for your supportive review and suggestions. Below we respond to the comments in **Weaknesses (W) and Questions (Q)**.
>
> ---
>
> ***W1: The primary weakness of this paper is the lack of in-depth analysis.***
>
> We encounter challenges in theoretically analyzing the behaviors of Transformers comprising multiple attention layers and MLP layers. What we could formulate theoretical interpretations is with repeated tokens as input in $\\textrm{\\color{blue}Table 1(Left)}$ (page 5), why GPT2 models still exhibit attention sink, whereas Mistral and LLaMA models do not. Please kindly review our response to ***Q3***. Regarding the KV bias section, we have included a more comprehensive explanation in the response to ***Q4***.
>
> ---
>
> ***Q1 (a): Performance curve of validation performance of each model against size?***
>
> Following your suggestions, we evaluate the performance of these pre-trained LMs on HellaSwag. $\\textrm{\\color{blue}Figure 10 (page 19)}$ visualizes the attention sink metric, accuracy, and normalized accuracy for comparative analysis. Within the same LM family, an increase in model scale correlates with improved downstream LM performance. However, within the OPT family and the GPT2 family, our attention sink metric indicates a similar level across different model scales. Besides, the OPT family exhibits more obvious attention sink compared to Pythia at comparable model scales, yet their downstream LM performance remains comparable.
>
> ---
>
> ***Q1 (b): Correlation between attention sink and validation loss?***
>
> The relationship between attention sink and validation loss seems to be not consistently positive or negative. In addition to the observations in $\\textrm{\\color{blue}Figure 10}$ (page 19), our controlled LM pretraining experiments indicate that, as exemplified by the weight decay in $\\textrm{\\color{blue}Table 2}$ (page 6), increased weight decay ratios result in a more pronounced attention sink, while the validation loss deteriorated after certain values.
>
> ---
>
> ***Q2: Fig. 4 (right) shows that a lower learning rate results in a slower increase in attention sink. Is it possible that attention sink occurs simply because the model has not been well-tuned?***
>
> We clarify that a lower learning rate not only results in a slower increase in attention sink, but also mitigates attention sink, even when we run more training steps. We conduct experiments by maintaining the constant product of the learning rate and training steps. $\\textrm{\\color{blue}Table 9}$ (page 25) indicates that LMs trained with lower learning rates and more steps still exhibit less obvious attention sink.
>
> There is a possibility that attention sink occurs because the model has not been well-tuned, which still remains unresolved. Optimization, specifically the learning rate, appears to substantially influence the attention sink. This suggests the potential existence of several optimal LM solutions, which exhibit no attention sink. The intriguing aspect is why mainstream optimization algorithms yield LM solutions with attention sink. As illustrated in $\\textrm{\\color{blue}Figure 4 (Left)}$ (page 5), all these mainstream pre-trained LMs exhibit attention sink without exceptions. This will be reserved for future endeavors.

---

> ### Author Response · Authors · 2024-11-19
> **Rebuttal by Authors [2/3]**
>
> ***Q3: In Table 1, GPT-XL behaves very differently from Llama and Mistral. Do you have any intuition as to why this might be?***
>
> Such different behaviors could be attributed to the positional embeddings (PE). GPT2-XL utilizes learnable PE, whereas Llama and Mistral adopt Rotary. In Appendix B.1, we theoretically demonstrate that for LMs utilizing NoPE/relative PE/ALiBI/Rotary, if the initial $T$ tokens are the same, their corresponding hidden states are also identical. Consequently, they all have massive activations, thus dispersing the attention sink. This explains the disappearance of attention sink in Llama/Mistral. Additionally, we derive the closed form/upper bound for attention scores in LMs utilizing NoPE/relative PE/ALiBI/Rotary via **Propositions 1-4 in Appendix B.1**.
>
> In LMs with learnable PE, despite the same word embeddings for repeated tokens, the PEs assigned to each token position differ, leading to distinct hidden states. Therefore, only the first token has massive activations. Then the initial token attention sink still exists. We incorporate new experiments showing that attention sink in GPT2-XL is strongly linked to the first PE vector $\\boldsymbol{p}\_1$. As shown in $\\textrm{\\color{blue}Table 7}$ (page 18), upon replacing $\\boldsymbol{p}_1$ with other vectors $\\boldsymbol{p}\_{t\\neq 1}$, the amplitude of attention sink on the first token is significantly diminished. When swapping the positions of $\\boldsymbol{p}\_1$ and $\\boldsymbol{p}\_{t\\neq 1}$, the $t$-th token becomes the new sink token.
>
> ---
>
> ***Q4: Could you provide some mathematical formulation on how the attention variants that excludes attention sink (2nd to 4th rows in Table 4) are correlated?***
>
> - The first row in $\\textrm{\\color{blue}Table 4}$ (page 9) illustrates the standard attention operation within a single head: $\\textrm{Softmax}\\left(\\frac{1}{\\sqrt{d\_h}}\\boldsymbol{Q}\^{l\\textrm{,}h}{\\boldsymbol{K}\^{l\\textrm{,}h}}\^\\top+ \\boldsymbol{M}\\right)\\boldsymbol{V}\^{l\\textrm{,}h}$. Here $\\boldsymbol{Q}\^{l\\textrm{,}h},\\,\\boldsymbol{K}\^{l\\textrm{,}h},\\,\\boldsymbol{V}\^{l\\textrm{,}h}\\in \\mathbb{R}\^{T\\times d\_h}$ denote to the qkv matrices for the $T$ tokens.
>
> - The 2nd row: $\\textrm{Softmax}\\left(\\frac{1}{\\sqrt{d\_h}}\\begin{bmatrix}
>    \\boldsymbol{q}\^{\*l\\textrm{,}\\,h} \\\\
>         \\boldsymbol{Q}\^{l\\textrm{,}\\,h}
>     \\end{bmatrix}\\begin{bmatrix}
>        {\\boldsymbol{k}\^{\*l\\textrm{,}\\,h}}\^\\top \& {\\boldsymbol{K}^{l\\textrm{,}\\,h}}\^\\top
>     \\end{bmatrix}+ \\boldsymbol{M}\\right)\\begin{bmatrix}
>    \\boldsymbol{v}\^{\*l\\textrm{,}\\,h} \\\\
>     \\boldsymbol{V}\^{l\\textrm{,}\\,h}
> \\end{bmatrix}$
>  denotes the incorporation of learnable qkv biases (sink token $\\boldsymbol{x}\^{\*}$), resulting in modified qkv matrices $\\boldsymbol{Q}’\^{l\\textrm{,}\\,h},\\,\\boldsymbol{K}’\^{l\\textrm{,}\\,h},\\,\\boldsymbol{V}’\^{l\\textrm{,}\\,h}\\in \\mathbb{R}\^{(T+1)\\times d\_h}$.  In this scenario, the sink token $\\boldsymbol{x}\^{\*}$ becomes the first token (visible to all other tokens) and absorbs the attention sink from the actual first token $\\boldsymbol{x}\_1$. Subsequently, a question arises: do we genuinely require the q biases $\\boldsymbol{q}\^{\*l\\textrm{,}\\,h}$ as only the k biases $\\boldsymbol{k}\^{\*l\\textrm{,}\\,h}$  and v biases $\\boldsymbol{v}\^{\*l,h}$ contribute to the calculation of attention scores for the subsequent tokens $\\boldsymbol{x}\_{1:T}$?
>
> - The above question motivates the 3rd row: $\\textrm{Softmax}\\left(\\frac{1}{\\sqrt{d\_h}}\\boldsymbol{Q}\^{l\\textrm{,}\\,h}\\begin{bmatrix}
>        {\\boldsymbol{k}\^{\*l\\textrm{,}\\,h}}\^\\top \& {\\boldsymbol{K}\^{l\\textrm{,}\\,h}}\^\\top
>     \\end{bmatrix}+ \\boldsymbol{M}\\right)\\begin{bmatrix}
>     \\boldsymbol{v}\^{\*l\\textrm{,}\\,h} \\\\
>     \\boldsymbol{V}\^{l\\textrm{,}\\,h}
> \\end{bmatrix}$. This makes the qkv matrices $\\boldsymbol{Q}’\^{l\\textrm{,}\\,h}\\in \\mathbb{R}\^{T\\times d\_h} ,\\,\\boldsymbol{K}’\^{l\\textrm{,}\\,h},\\,\\boldsymbol{V}’\^{l\\textrm{,}\\,h}\\in \\mathbb{R}\^{(T+1)\\times d\_h}$. In this scenario, although there is no explicit sink token, the k bias $\\boldsymbol{k}^{\*l,h}$ is present in the first position and absorbs the attention sink.
>
> - Finally, the 4th row has a similar form as the 3rd row. It indicates that, despite a v bias of all zeros, the k bias $\\boldsymbol{k}\^{\*l\\textrm{,}\\,h}$ can still absorb the attention sink.
>
> To summarize, the attention variants in 2nd to 4th row in $\\textrm{\\color{blue}Table 4}$ (page 9) introduce additional biases to the qkv matrices. Crucially, attention sink occurs in the first k, irrespective of its origin (the actual first token $\\boldsymbol{x}\_1$, an added sink token $\\boldsymbol{x}\^{\*}$, or introduced bias $\\boldsymbol{k}\^{\*}$). This represents the intrinsic relationship among these attention variants.

---

> ### Author Response · Authors · 2024-11-19
> **Rebuttal by Authors [3/3]**
>
> ***Q5: In Table 6, it appears that the normalizer may impact the attention sink. Do you have additional evidence with other types of normalizers?***
>
> In $\\textrm{\\color{blue}Table 14 (Left)}$ (page 28), we have previously examined the scaling of normalization, i.e., making the attention scores sum to $\\alpha<1$.  This leads to mitigated attention sink. In the revision, we additionally consider a normalizer $\\boldsymbol{Z}\_i=[\\sum\_{j'=1}\^i\\textrm{sim}(\\varphi(\\boldsymbol{q}\_i)\\textrm{,}\\,\\varphi(\\boldsymbol{k}\_{j'}))\^p]\^{\\frac{1}{p}}$, which makes power $p$ of attention scores sum to one.
>
> For softmax attention, given $\\textrm{sim}(\\varphi(\\boldsymbol{q}\_i)\\textrm{,}\\,\\varphi(\\boldsymbol{k}\_j))=\\textrm{exp}(\\frac{\\boldsymbol{q}\_i\^\\top\\boldsymbol{k}\_j}{\\sqrt{d\_h}})$, we have
> $\\boldsymbol{v}\_i\^{\\dagger}=\\frac{\\sum\_{j=1}\^i\\textrm{exp}(\\frac{\\boldsymbol{q}\_i\^\\top\\boldsymbol{k}\_j}{\\sqrt{d\_h}}))\\boldsymbol{v}\_j}{\\left(\\sum\_{j'=1}\^i\\textrm{exp}(\\frac{\\boldsymbol{q}\_i\^\\top\\boldsymbol{k}\_{j'}}{\\sqrt{d\_h}})\^p\\right)\^{\\frac{1}{p}}}=\\sum\_{j=1}\^i \\left(\\frac{\\textrm{exp}(\\frac{\\boldsymbol{q}\_i\^\\top\\boldsymbol{k}\_j}{\\sqrt{d\_h}/p})}{\\sum\_{j'=1}\^i\\textrm{exp}(\\frac{\\boldsymbol{q}\_i\^\\top\\boldsymbol{k}\_{j'}}{\\sqrt{d\_h}/p})}\\right)\^{\\frac{1}{p}}\\boldsymbol{v}\_j$. This is equivalent to incorporating a temperature $1/p$ into the softmax attention logits, followed by extracting the $p$-root of attention scores after softmax. This is referred to as $p$-normalized softmax attention. $p=\\textrm{1}$ in standard softmax attention. Similarly, we develop the LMs utilizing $p$-normalized sigmoid attention.
>
> For LMs with $p$-normalized softmax attention, we find when $p=2$ or $p=3$ or $p=4$, the pre-training diverges, resulting in an infinite loss. When $p=1/2$, $p=1/3$ or $p=1/4$, the pre-training converges. Since the attention scores do not sum to one, we examine the massive activations instead, as depicted in $\\textrm{\\color{blue}Figure 29 (Left)}$ (page 29). It is observed that smaller values of $p$ mitigate massive activations, yet they are less effective than sigmoid attention without normalization. To intuitively understand this, a smaller $p$ results in a higher temperature during the softmax operation, leading to more flattened attention logits.
>
> For LMs with $p$-normalized sigmoid attention, there is no training diverging problem when $p>1$. As depicted in $\\textrm{\\color{blue}Figure 29 (Right)}$ (page 29), trained LMs continue to exhibit massive activations. In conclusion, various normalizers can influence the amplitude of the attention sink and, in some cases, result in mitigated attention sink.

---

> > ### Comment · Reviewer_xfDZ · 2024-11-23
> >
> > Thank you for your thorough and detailed response. I appreciate the effort and clarity you’ve put into addressing the comments. This is an excellent study, and I continue to view it very positively. After reviewing your rebuttal and other reviews, I am maintaining my score of 8 and recommending for acceptance.

---

> > > ### Author Response · Authors · 2024-11-23
> > > **Thank you for your support**
> > >
> > > We appreciate your detailed feedback and suggestions, which greatly help us to improve our work! In the final revision, we will incorporate the new empirical results and derivations to further improve our paper. Thank you again!

---

### Official Review · Reviewer_5eYD · 2024-11-04

**Soundness:** 4
**Presentation:** 3
**Contribution:** 3
**Rating:** 6
**Confidence:** 4

**Summary:**

The paper investigates the phenomenon of attention sink in LMs, where significant attention is allocated to the first token regardless of its semantic importance. Key findings include the correlation of the sink position with the loss function and data distribution, and the behavior of attention sink as key biases storing extra attention without contributing to value computation. The paper also shows that attention sinks do not emerge when softmax attention is replaced with other operations like sigmoid attention without normalization, up to 1B parameters.

**Strengths:**

- The paper provides a thorough investigation into the attention sink phenomenon, covering various factors that influence its emergence in LMs.
- The findings have practical applications in areas such as streaming/long context generation, KV cache optimization, and model quantization.
- The study considers various model architectures, optimizers, and data distributions, offering a broad perspective on the attention sink phenomenon.

**Weaknesses:**

- The study notes that attention sink disappears with less training data, suggesting that the models might be overfitting to certain aspects of the training data. It seems that these two factors cannot be decoupled, and it is impossible to explain whether it is over-fit or the amount of data that affects attention-sink.
- The study claims that it focuses on the Language Models. However, this work pay more attention on auto-regressive LMs and may not capture the nuances of other types of encoder-like LMs or Jamba. Does those architecture can solve the attention-sink phenomenon?

**Questions:**

- How to identify the token where attention-sink occurs? How to determine those locations?

---

> ### Author Response · Authors · 2024-11-19
> **Rebuttal by Authors**
>
> Thank you for your supportive review and suggestions. Below we respond to the comments in **Weaknesses (W)** and **Questions (Q)**.
>
> ---
>
> ***W1: It is impossible to explain whether it is over-fit or the amount of data that affects attention sink.***
>
> To investigate this issue further, we monitor the training dynamics of train/validation loss and our attention sink metric in configurations with limited training data, specifically, 50M and 100M.  We additionally assess the results utilizing the default configuration of 5B training data for reference. All these findings are depicted in $\\textrm{\\color{blue}Figure 28}$ (page 25).
>
> Our findings indicate that with merely 50M and 100M training data, LMs exhibit overfitting at initial phases, specifically between 1k and 2k steps. Simultaneously, $\\textrm{Sink}\_1\^{\\epsilon}$ keeps an exceedingly minimal value (below 1\%). In the setup of the 5B training data, $\\textrm{Sink}\_1\^{\\epsilon}$ continues to rise after a specific step. This suggests that *the amount of training data, rather than overfitting, significantly influences the emergence of attention sink*.
>
> ---
>
> ***W2: Do encoder-like LMs and Jamba can solve the attention-sink phenomenon?***
>
> This is an insightful question! However, **architecture may not solve the attention-sink phenomenon**, as explained below:
>
> - In encoder-only Transformers, a similar phenomenon of “attention sink” also occurs. As demonstrated in [1], BERT also assigns significant attention scores to the [SEP] token, which functions as the sink token. Furthermore, as elucidated in [2], artifacts (patch tokens) are detected in the attention maps of vision transformers (encoder-only transformers). These artifacts absorb a significant amount of attention, referred to as “registers”, analogous to “attention sink” in [1]. Unlike sink tokens in auto-regressive LMs, these registers do not consistently appear as the initial token and hold global information. Our experiments indicate that sink tokens in auto-regressive LMs possess negligible or no semantic meaning. We posit that the softmax operation also significantly contributes to the aforementioned attention sink phenomenon, even within encoder-only Transformers.
>
> - As you suggested, we incorporate additional experiments to measure Jamba’s attention sink utilizing similar methodologies as presented in our paper. Both Jamba-v0.1 [3] and Jamba-1.5 Mini [4] consist of 32 layers, which include 4 transformer layers. Initially, our findings show that $\\textrm{Sink}\_1\^{\\epsilon}=\\textrm{88.48}\\%$ for Jamba-v0.1 and $\\textrm{Sink}\_1\^{\\epsilon}=\\textrm{87.88}\\%$ for Jamba-1.5 Mini, indicating a strong attention sink on the first token. Subsequently, we include multiple visualizations of the attention sink in Jamba-v0.1 and  Jamba-1.5 Mini in $\\textrm{\\color{blue}Figure 25-27}$ (page 23-24). It is noted that the majority of heads exhibit an obvious attention sink, except for a few heads in the third Transformer layer.
>
> In the final revision, we will conduct more investigations on different LM architectures to further validate our conclusions.
>
> ---
>
> ***Q1: How to identify the token where attention-sink occurs? How to determine those locations?***
>
> We present our threshold-based attention sink metric in Section 3.2: $\\textrm{Sink}\_k\^{\\epsilon}=\\frac{1}{L}\\sum\_{l=1}\^L\\frac{1}{H}\\sum\_{h=1}\^H\\mathbb{I}(\\alpha\_k\^{l\\textrm{,}h}>\\epsilon)$. This metric can also identify the location of where attention-sink occurs. If $\\textrm{Sink}\_k\^{\\epsilon}$ is significantly larger than 0, we could regard $k$-th token as a sink token.
>
> From an alternative viewpoint, [5] showed that attention sink is strongly correlated with massive activations. The hidden states $\\boldsymbol{h}\_k\^l$ of the sink token exhibit a significantly larger $\\ell\_2$-norm compared to those of other tokens. Consequently, it is logical to employ the ratio of $\\ell\_{2}$-norm of hidden states to the mean or median values for the identification of the sink token: $\\textrm{Sink}’\_k=\\frac{1}{L}\\sum\_{l=1}^L\\frac{||\\boldsymbol{h}\_k\^{l}||\_2}{\\textrm{mean}\_{1\\leq t \\leq T}(||\\boldsymbol{h}\_t\^{l}||\_2)}$ (the mean operator could be substituted with the median). If  $\\textrm{Sink}’\_k$ is significantly larger than 1, we consider $k$-th token a sink token.
>
> ---
>
> ***References:*** \
> [1] Xiao et al. Efficient streaming language models with attention sinks. ICLR 2024\
> [2] Darcet et al. Vision Transformers Need Registers. ICLR 2024\
> [3] Jamba: A Hybrid Transformer-Mamba Language Model. Arxiv 2024\
> [4] Jamba-1.5: Hybrid Transformer-Mamba Models at Scale. Arxiv 2024\
> [5] Sun et al. Massive activations in large language models. COLM 2024

---

### Author Response · Authors · 2024-11-19
**Summary of Paper Revision**

We thank all reviewers for their constructive feedback, and we have responded to each reviewer individually. We have also uploaded a **Paper Revision** including additional results and illustrations:

- $\\textrm{\\color{blue}Table 7}$ (page 18): new experiments that demonstrate how learnable positional embeddings affect the property of attention sink;
- $\\textrm{\\color{blue}Figure 10}$ (page 19): relation of attention sink and LM performance;
- $\\textrm{\\color{blue}Figure 11-14}$ (page 19): more visualizations of $\\ell\_2$-norm of hidden states/keys/values;
- $\\textrm{\\color{blue}Figure 15-18}$ (page 20-21): more visualizations of QK angles;
- $\\textrm{\\color{blue}Figure 19-24}$ (page 21-23): visualizations of block-wise and head-wise distributions of attention sink;
- $\\textrm{\\color{blue}Figure 25-27}$  (page 23-24): visualizations and discussions of attention sink in Jamba models;
- $\\textrm{\\color{blue}Table 9}$ (page 25): new experiments to demonstrate that attention sink is less obvious in LMs trained with small learning rates even after accounting for more training steps;
- $\\textrm{\\color{blue}Figure 28}$ (page 25): new experiments to show that small training data amount, rather than overfitting, leads to the disappearance of attention sink;
- $\\textrm{\\color{blue}Table 13}$ (page 27): new experiments that demonstrate the sink token’s key is distributed in a different manifold with a low rank;
- $\\textrm{\\color{blue}page 27-28}$: use mathematical formulations to transform attention score re-scaling into a scenario where we only rescale the initialization and learning rate of $\\boldsymbol{W}\_O$ or $\\boldsymbol{W}\_V$ under the Adam/AdamW optimizer;
- $\\textrm{\\color{blue}Figure 29}$ (page 29): new experiments to show the effects of normalizers in attention operations on attention sink;
- $\\textrm{\\color{blue}Figure 30}$ (page 30): new experiments demonstrating LMs with sigmoid attention without normalization have no issues of training stability during supervised fine-tuning.

---

### Author Response · Authors · 2024-11-22
**Looking forward to further feedback**

Dear Reviewers,

Thank you again for your valuable comments and suggestions, which are really helpful for us. We have posted responses to the proposed concerns and included additional experiment results.

We totally understand that this is quite a busy period, so we deeply appreciate it if you could take some time to return further feedback on whether our responses solve your concerns. If there are any other comments, we will try our best to address them.

Best,

The Authors

---

### Meta-Review · Area_Chair_wV4Y · 2024-12-25

**Metareview:**

This paper explores the phenomenon of attention sink in language models (LMs). Attention sink describes how, in autoregressive Transformer-based LMs, a disproportionate amount of attention often gets allocated to the first token in the sequence, regardless of its semantic importance. The authors provide extensive empirical evidence that attention sink arises across model sizes and architectures, and sufficient training data and high learning rates facilitate the emergence of attention sink. The author also showed the root of attention sink is from softmax, which can be greatly prevented by other attention variants (e.g., sigmoid attention).

**Strengths** (1) All reviewers agree on the paper's breadth of experiments across various training conditions, architectures and hyperparameters. (2) Understanding how attention sink works is also crucial for various LLM applications

**Weaknesses** (1) One recurring critique is that certain parts of the study feel more like empirical observations. A mathematical explanation for why specific attention variants eliminate attention sink would be preferred; (2) The reviewers would like to see more evidence on whether attention sink harms or helps real downstream performance.

**Decision** The paper’s strengths, in particular its extensive empirical scope, practical insights, and potential impact, makes it a clear acceptance.

**Additional Comments On Reviewer Discussion:**

The reviewers asked questions about how the attention-sink token is identified, how the hyperparameters take effect, and how the attention-sink can be mitigated. The authors responded accordingly and also pointed out future research directions on attention-sink. Overall the reviewers are satisfied with the discussion.

---

### Decision · Program_Chairs · 2025-01-22

Accept (Spotlight)